# Molecular mechanism for direct actin force-sensing by α-catenin

Lin Mei[1,2], Santiago Espinosa de los Reyes[1], Matthew J Reynolds[1], Rachel Leicher[2,3], Shixin Liu[3], Gregory M Alushin[1]*

[1]Laboratory of Structural Biophysics and Mechanobiology, The Rockefeller University, New York, United States; [2]Tri-Institutional PhD Program in Chemical Biology, The Rockefeller University, New York, United States; [3]Laboratory of Nanoscale Biophysics and Biochemistry, The Rockefeller University, New York, United States

**Abstract** The actin cytoskeleton mediates mechanical coupling between cells and their tissue microenvironments. The architecture and composition of actin networks are modulated by force; however, it is unclear how interactions between actin filaments (F-actin) and associated proteins are mechanically regulated. Here we employ both optical trapping and biochemical reconstitution with myosin motor proteins to show single piconewton forces applied solely to F-actin enhance binding by the human version of the essential cell-cell adhesion protein αE-catenin but not its homolog vinculin. Cryo-electron microscopy structures of both proteins bound to F-actin reveal unique rearrangements that facilitate their flexible C-termini refolding to engage distinct interfaces. Truncating α-catenin's C-terminus eliminates force-activated F-actin binding, and addition of this motif to vinculin confers force-activated binding, demonstrating that α-catenin's C-terminus is a modular detector of F-actin tension. Our studies establish that piconewton force on F-actin can enhance partner binding, which we propose mechanically regulates cellular adhesion through α-catenin.

*For correspondence:
galushin@rockefeller.edu

Competing interests: The authors declare that no competing interests exist.

## Introduction

Cells probe and respond to the mechanical properties of their surroundings through cytoskeletal networks composed of actin filaments (F-actin), myosin motor proteins, and dozens of actin-binding proteins (ABPs). These networks transmit forces through cell-matrix focal adhesions (*Humphrey et al., 2014*) and cell-cell adherens junctions (*Charras and Yap, 2018*), plasma membrane-associated many-protein assemblies which serve as hubs for the conversion of mechanical cues and stimuli into biochemical signaling cascades (mechanotransduction). Defects in mechanotransduction are associated with numerous diseases (*Jaalouk and Lammerding, 2009*), including muscular dystrophies, cardiomyopathies, and metastatic cancer, yet therapeutics which specifically target these pathways are largely absent due to our ignorance of the mechanisms that transduce mechanical signals through the cytoskeleton.

Diverse force regimes modulate the polymerization dynamics, micron-scale architecture, and protein composition of actin assemblies by molecular mechanisms that remain unclear (*Harris et al., 2018*; *Romet-Lemonne and Jégou, 2013*). Cellular-scale pressures in the range of hundreds of pascal regulate the assembly and mechanical power of branched-actin networks generated by the ARP2/3 complex in vitro (*Bieling et al., 2016*) and in vivo (*Mueller et al., 2017*), tuning network geometric properties including filament length, density, and distribution of filament orientations. Live-cell imaging studies identified a subset of ABPs that preferentially localize to the cytoskeleton in vivo in response to this magnitude of mechanical stimulation (*Schiffhauer et al., 2016*), which were also postulated to recognize properties of network geometry. Molecular-scale forces in the

**eLife digest** All of the cells in our bodies rely on cues from their surrounding environment to alter their behavior. As well sending each other chemical signals, such as hormones, cells can also detect pressure and physical forces applied by the cells around them. These physical interactions are coordinated by a network of proteins called the cytoskeleton, which provide the internal scaffold that maintains a cell's shape. However, it is not well understood how forces transmitted through the cytoskeleton are converted into mechanical signals that control cell behavior.

The cytoskeleton is primarily made up protein filaments called actin, which are frequently under tension from external and internal forces that push and pull on the cell. Many proteins bind directly to actin, including adhesion proteins that allow the cell to 'stick' to its surroundings. One possibility is that when actin filaments feel tension, they convert this into a mechanical signal by altering how they bind to other proteins.

To test this theory, Mei et al. isolated and studied an adhesion protein called α-catenin which is known to interact with actin. This revealed that when tiny forces – similar to the amount cells experience in the body – were applied to actin filaments, this caused α-catenin and actin to bind together more strongly. However, applying the same level of physical force did not alter how well actin bound to a similar adhesion protein called vinculin. Further experiments showed that this was due to differences in a small, flexible region found on both proteins. Manipulating this region revealed that it helps α-catenin attach to actin when a force is present, and was thus named a 'force detector'.

Proteins that bind to actin are essential in all animals, making it likely that force detectors are a common mechanism. Scientists can now use this discovery to identify and manipulate force detectors in other proteins across different cells and animals. This may help to develop drugs that target the mechanical signaling process, although this will require further understanding of how force detectors work at the molecular level.

piconewton range have also been shown to modulate the activity of F-actin polymerization (*Courtemanche et al., 2013*; *Jégou et al., 2013*; *Risca et al., 2012*; *Zimmermann et al., 2017*) and severing (*Hayakawa et al., 2011*) factors in vitro, suggesting that molecular components of actin networks could be mechanically regulated by physiological forces. While F-actin binding by the actin-depolymerization factor cofilin was reported to be directly regulated by tension across the filament to modulate its activity (*Hayakawa et al., 2011*), a recent study has suggested that cofilin is tension-insensitive, instead having its severing activity regulated by filament bending (*Wioland et al., 2019*). F-actin has been reported to adopt a structural landscape of co-existing conformations in cryo-electron microscopy (cryo-EM) studies (*Galkin et al., 2010b*), leading to speculation that actin filaments could themselves serve as tension sensors by presenting distinct binding interfaces to ABPs in the presence of load (*Galkin et al., 2012*). It nevertheless remains unclear to what extent mechanical modulation of functional interactions between ABPs and F-actin occurs through direct regulation of F-actin's binding interactions by force. Furthermore, structural mechanisms enabling ABPs to detect force on F-actin, to our knowledge, are unknown.

Inspired by the report that the enhanced cytoskeletal localization response of ABP isoforms differed substantially when cells were mechanically stimulated (*Schiffhauer et al., 2016*), we reasoned that biophysical and structural analysis of closely-related ABPs would be a powerful approach for mechanistically dissecting mechanically regulated F-actin binding. Here we specifically investigate enhanced binding to F-actin when the load is applied solely to the filament, which we refer to as 'force-activated binding'. Our studies focus on the homologous adhesion proteins α-catenin (*Kobielak and Fuchs, 2004*) and vinculin (*Ziegler et al., 2006*), which are major ABP components found in adherens junctions (α-catenin and vinculin) and focal adhesions (vinculin) that are critical for the force-dependent strengthening of cellular adhesion and mechanotransduction (*Dumbauld et al., 2013*; *Yonemura et al., 2010*). Vinculin is a strictly auto-inhibited globular protein that must engage with multiple adhesion partners to be activated and bind F-actin (*Johnson and Craig, 1995*), stabilizing adhesion through incompletely defined mechanisms. On the other hand, α-catenin exists in two distinct populations maintained in dynamic equilibrium in the cell (*Drees et al., 2005*). It serves as a

central component of the membrane-anchored heterotrimeric α-catenin–β-catenin–cadherin complex (the 'cadherin-catenin complex') at adherens junctions, which lacks F-actin-binding activity in traditional assays when isolated (*Yamada et al., 2005*). It also forms a soluble homodimer with constitutive modest F-actin-binding activity, thought to play a role in the generation and maintenance of actin bundles by cross-linking filaments and inhibiting ARP2/3 binding, thereby suppressing branched-actin formation (*Drees et al., 2005*). The structural mechanisms by which forces transmitted through the actin cytoskeleton modulate the complex networks of binding interactions formed by α-catenin and vinculin during adhesion maturation remain unknown.

Both proteins are entirely α-helical (*Bakolitsa et al., 2004*; *Rangarajan and Izard, 2013*) and are composed of a large N-terminal 'head' domain, which engages in protein-protein interactions with other adhesion molecules (*Kobielak and Fuchs, 2004*; *Ziegler et al., 2006*) and a smaller C-terminal 5-helix bundle F-actin binding 'tail' domain (*Bakolitsa et al., 1999*; *Ishiyama et al., 2013*; *Figure 1— figure supplement 1*, Helices H1–H5), connected by a flexible linker. The isolated actin-binding domains (ABDs), which we utilize in our study, retain their structures and actin-binding activities (*Bakolitsa et al., 1999*; *Ishiyama et al., 2013*), engaging a similar site on the filament surface (*Hansen et al., 2013*; *Janssen et al., 2006*; *Kim et al., 2016*; *Thompson et al., 2014*). Recent single-molecule force-spectroscopy studies reported that both α-catenin in the cadherin-catenin complex (*Buckley et al., 2014*) and vinculin (*Huang et al., 2017*) form catch bonds with F-actin, characterized by increased bond lifetime when moderate forces around 10 pN are applied across the ABP-actin interface. These studies provided one potential resolution to the apparent contradiction between biochemical studies demonstrating the cadherin-associated α-catenin population lacks F-actin binding activity (*Drees et al., 2005*) and cellular studies suggesting α-catenin plays a key role in linking adherens junctions to the cytoskeleton (*Kobielak and Fuchs, 2004*). However, in vivo both vinculin and α-catenin primarily engage contractile actomyosin bundles, whose component actin filaments are constitutively exposed to myosin-generated forces. This led us to hypothesize that force-activated binding to tensed F-actin could also be a key regulatory mechanism at adhesions, suitable for promoting the formation of initial attachments to actomyosin through membrane-anchored ABPs (e.g. the cadherin complex) before the ABP-F-actin interface itself coming under load. Furthermore, it could also serve as a mechanism to enrich soluble ABPs (e.g. the α-catenin dimer) whose binding geometry is fundamentally incompatible with catch-bond formation.

Here using simultaneous optical trapping and confocal fluorescence microscopy, we show that tension on the order of 1 pN across actin filaments directly enhances F-actin binding by human αE-catenin, but not vinculin. Utilizing a novel Total Internal Reflection Fluorescence (TIRF) microscopy in vitro reconstitution assay, we further show that physiological forces generated by myosin motor proteins activate α-catenin F-actin binding. Approximately 3 Å resolution cryo-EM structures of both proteins bound to F-actin (to our knowledge, the highest-resolution structures of ABP-F-actin complexes reported to date) reveal they share an overlapping major actin-binding site. However, they undergo markedly different rearrangements at their flexible N- and C-termini upon binding, resulting in distinct contacts mediated by their C-terminal extensions (CTEs) which re-fold on the actin surface. Truncating α-catenin's CTE results in constitutive strong F-actin binding regardless of force, and a chimeric construct of vinculin featuring α-catenin's CTE gains force-activated binding, demonstrating the α-catenin CTE to be a modular 'force-detector' which negatively regulates low-force binding. Together, our studies indicate piconewton force on F-actin can be discriminated by flexible elements in ABPs to mediate direct force-activated binding.

## Results

### Piconewton tension across individual actin filaments directly activates binding by α-catenin

To determine whether the actin-binding activity of vinculin or α-catenin could be regulated solely by force across F-actin, we performed correlative force and fluorescence measurements with a commercial instrument (*Hashemi Shabestari et al., 2017*) that combines dual-trap optical tweezers and confocal fluorescence microscopy (*Figure 1A*). In these experiments, Alexa-555 phalloidin-stabilized biotinylated actin filaments were captured from a laminar stream across a microfluidic flow-cell between two optically trapped streptavidin-coated beads. Tethered filaments were then transferred

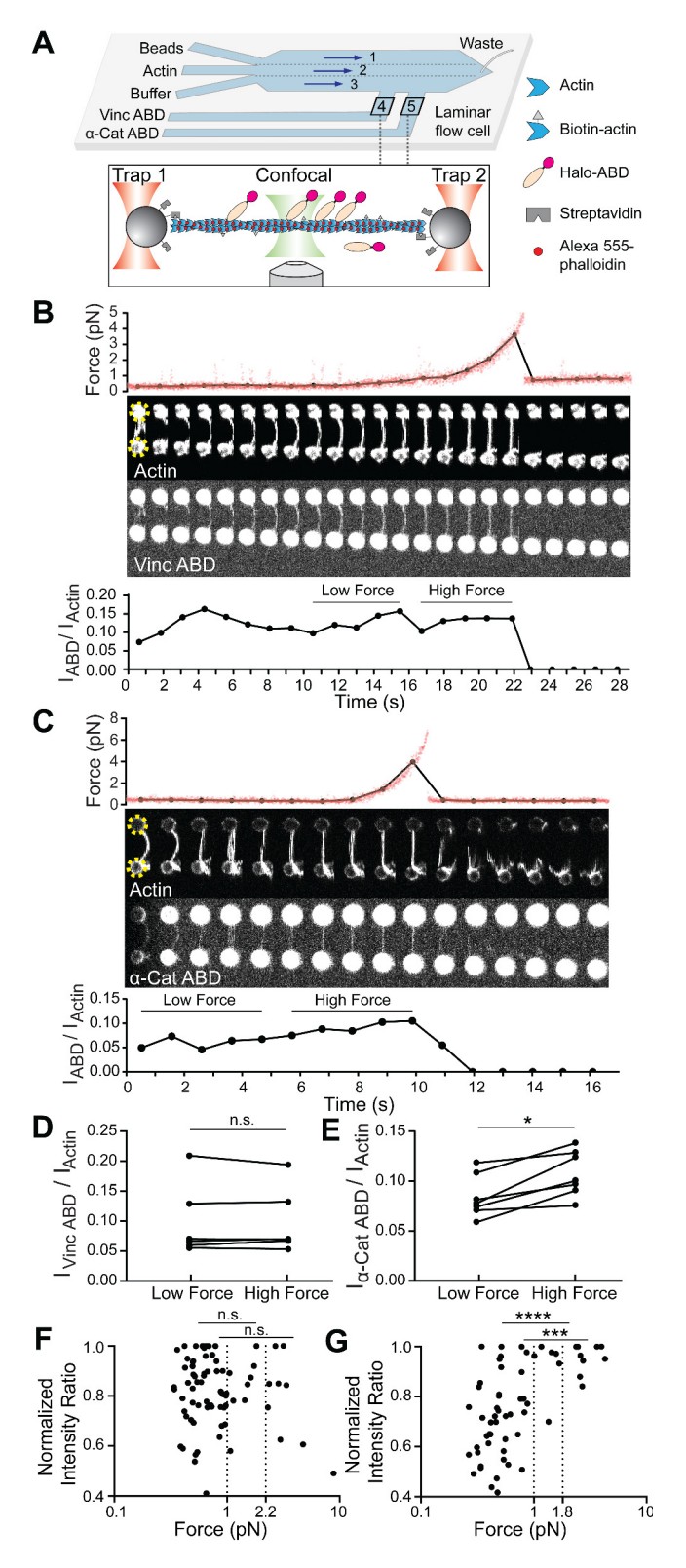

**Figure 1.** Piconewton load across individual filaments activates α-catenin binding, while vinculin is force insensitive. (**A**) Top: Geometry of the microfluidic chamber. Bottom: Cartoon of correlative single-filament force and fluorescence assay. (**B and C**) Representative single-filament constant velocity pulling experiments (0.1 µm/s) for vinculin ABD (**B**) and α-catenin ABD (**C**). Top: Force versus time plot. Red dots, raw data; black dots, force

*Figure 1 continued on next page*

*Figure 1 continued*

values binned to an interval of confocal frames. Middle: Montage of confocal frames of actin and ABD channels. Positions of beads are indicated with yellow circles (diameter, 4 µm). Bottom: $I_{ABD}/I_{actin}$ ratio versus time plot. High-force: Average of five intensity ratios before actin breakage; low-force: average of five intensity ratios immediately before high-force average. Concentration of vinculin ABD or α-catenin ABD: 2 µM. (D and E) Paired analysis of low-force and high-force averages across trials for vinculin ABD (D) (N = 6, p=0.84) and α-catenin ABD (E) (N = 7, p=0.0156). Wilcoxon signed-rank test: *p<0.05; n.s. (not significant), p>0.05. (F and G) Scatter plots of normalized fluorescence ratios versus force. Statistical comparisons are between intensity ratios divided into two bins: force below and above 1 pN or 2.2 pN (dotted lines) for vinculin ABD (F) (1 pN: below: n = 62, above: n = 20, p=0.65; 2.2 pN: below: n = 72, above: n = 10, p=0.74), and force below and above 1 pN or 1.8 pN (dotted lines) for α-catenin ABD (G) (1pN: below: n = 53, above: n = 15, p<0.0001; 1.8 pN: below: n = 59, above: n = 9, p=0.0002). KS test: ****p<0.0001; ***p<0.001; n.s., p>0.05.

The online version of this article includes the following figure supplement(s) for figure 1:

**Figure supplement 1.** Multiple sequence alignments of human vinculin and α-catenin isoforms.
**Figure supplement 2.** Additional analysis of single filament correlative force and fluorescence assays.

to a reservoir containing Halo-tagged ABD labeled with Alexa-488 and subjected to a constant velocity pulling protocol while simultaneously recording confocal movies of both the actin and ABD fluorescence signals.

We frequently observed a monotonic increase in force once a threshold extension was reached, followed by an instantaneous return to baseline (*Figure 1B,C*; *Figure 1—figure supplement 2A,B*), consistent with resistance from a tether composed of one or more actin filaments extended beyond their resting length (visible as straightening in the actin fluorescence channel) followed by force-induced breakage. To correlate force with ABD binding, we calculated the background-subtracted ratio of the ABD fluorescence intensity to the actin intensity (which we refer to as '$I_{ABD}/I_{actin}$') in each frame of the confocal movies (Materials and methods; *Figure 1B,C*, bottom), as well as the corresponding average force during frame acquisition (*Figure 1B,C*, top, black connected points). Apparent binding by the vinculin ABD fluctuated and did not change in response to load during individual pulling trajectories (*Figure 1B*), consistent with a previous report that mechanical stimulation does not promote cytoskeletal localization of vinculin in cells (*Schiffhauer et al., 2016*). However, we observed an apparent steady increase in binding along individual tethers by the α-catenin ABD (*Figure 1C*), consistent with force-activated binding. To quantify this phenomenon, due to the inherent fluctuations of individual $I_{ABD}/I_{actin}$ traces in this assay (potentially due to dynamic changes in ABD binding density along individual tethers and instability in the focus during confocal imaging), we first focused our analysis on a subset of long-lived tethers which featured at least 10 frames. We examined the difference between a 'high-force' average, defined as the mean $I_{ABD}/I_{actin}$ value from the five frames before the final tether rupture, and a 'low-force' average, defined as the mean $I_{ABD}/I_{actin}$ value from the five preceding frames during each recording (*Figure 1B,C*). Paired analysis of low-force/high force averages (Materials and methods) showed a significant increase consistent with force-activated binding only by the α-catenin ABD (*Figure 1D and E*; *Figure 1—figure supplement 2A*).

Next, we sought to determine the impacts of force on ABD binding at the level of individual actin filaments. As our tether assembly procedure captures filaments from solution, in principle each tether could be composed of a single filament or multiple filaments. We thus elaborated the possible multi-filament tether configurations and designed a series of criteria to identify and exclude them from the analysis. The first plausible multi-filament tether configuration (1) is composed of multiple filaments of different lengths, all attached to the trapped beads at both ends (*Figure 1—figure supplement 2B*), which we identified by the presence of multiple breaking peaks in the force curve, generated as the shortest remaining filament in the tether reaches full extension then ruptures (*Figure 1—figure supplement 2B*). The next plausible configuration (2) is multiple filaments of different lengths, not all of which are attached at both ends (*Figure 1—figure supplement 2C*), which we identified by fluorescence line scans of the actin intensity, where we observe step-like reductions which we interpret to correspond to the ends of non-bridging filaments (*Figure 1—figure supplement 2C*). The third and final configuration (3), composed of multiple filaments, all of essentially

identical length and attached at both ends (*Figure 1—figure supplement 2D*), is the most difficult to rigorously identify. We observed significant heterogeneity in the raw actin fluorescence intensity values between tethers (potentially due to slight differences in focus), with no significant difference between configuration one tethers and those featuring a single rupture force peak (data not shown), suggesting raw actin intensity cannot be used as a proxy for filament number. We thus analyzed the distribution of final breaking forces for all trials (pooling data collected in the presence of both the α-catenin and vinculin ABD), reasoning that configuration three tethers composed of multiple fully-extended actin filaments in parallel at the final break would generally rupture at higher forces. Consistent with this prediction, we observed a single major peak in the breaking force distribution centered at ~6 pN, which is likely to primarily consist of single filaments, as well as a long tail of $\geq$16.5 pN breaking forces, which we interpret to primarily encompass configuration three tethers. We thus additionally excluded all tethers with $\geq$16.5 pN breaking force from further analysis.

Based on these criteria, only 21 tethers (11 vinculin ABD and 10 α-catenin ABD) out of 183 total trials were included. This subset very likely predominantly contains tethers composed of single filaments. Consistent with the expected relative fragility of single-filament tethers with low breaking forces, most of these recordings only contained a limited number of frames, precluding quantification of force-dependent binding changes in individual tethers. We thus instead examined the relationship between force magnitude and ABD binding by pooling our recordings for each protein and plotting normalized $I_{ABD}/I_{actin}$ versus force (*Figure 1F,G*). This analysis revealed no apparent correlation between vinculin ABD binding and force (*Figure 1F*). However, the α-catenin ABD plot showed an apparent step-like transition from force-uncorrelated binding to consistent strong binding above a threshold force of approximately 1 pN (*Figure 1G*).

To quantify this phenomenon, we initially examined the distribution of intensity ratios above and below the subjectively identified 1 pN threshold, finding a significant difference only for the α-catenin ABD (*Figure 1F,G*). To objectively define the threshold force, we performed K-means clustering analysis (Materials and methods) which revealed that the α-catenin ABD force-fluorescence distribution could be optimally divided into two clusters with a threshold force of 1.8 pN. We found that the intensity ratios were also significantly different between these clusters (*Figure 1G*). Although there was no obvious correlation between force and binding by the vinculin ABD, as a control we nevertheless used K-means to divide the data into two clusters, which separated at a threshold force of 2.2 pN. Consistent with the lack of a force-binding correlation for the vinculin ABD, we find no significant difference between the distribution of intensity ratios in these two clusters (*Figure 1F*). This analysis, which is insensitive to the exact force threshold employed, supports a force-dependent increase in F-actin binding only for the α-catenin ABD. Collectively, these data suggest that piconewton-level tensile force along individual actin filaments is sufficient to activate α-catenin's F-actin binding.

## Physiological forces generated by myosin motors activate F-actin binding by α-catenin

As ~1 pN is the magnitude of force generated by individual myosin motor domains (*Finer et al., 1994*), we hypothesized α-catenin's F-actin binding would also be enhanced by physiological forces generated by myosins. To test this hypothesis, we developed a novel adaptation of the gliding filament assay (*Kron and Spudich, 1986*) to apply force to filaments mimicking actomyosin contractility in vivo (*Figure 2A*). In our preparation, plus-end directed myosin V motor proteins and minus-end directed myosin VI motor proteins are randomly surface-immobilized inside a flow chamber assembled on a cover glass for TIRF, resulting in a configuration where the motors are poised to engage in tug-of-war along non-stabilized, rhodamine-labeled actin filaments. A Halo-tagged, JF-646 (*Grimm et al., 2015*) labeled ABD is then flowed into the chamber in the absence of ATP, and a 2-color TIRF movie is recorded to visualize the basal level of actin binding when filaments are anchored by the rigor-state motors. The ABD is then re-introduced into the same chamber in the presence of ATP to activate the motors, and a second movie is recorded to visualize binding in the presence of force generation. Visual inspection of -ATP/+ATP TIRF movie pairs for the vinculin and α-catenin ABDs suggest they respond to motor-generated forces on F-actin distinctly. Actin localization of the vinculin ABD did not change in response to motor activity (*Figure 2B*; *Figure 2—video 1*); however, α-catenin ABD's actin localization was enhanced upon motor activation, indicative of force-activated actin binding (*Figure 2C*; *Figure 2—video 2*).

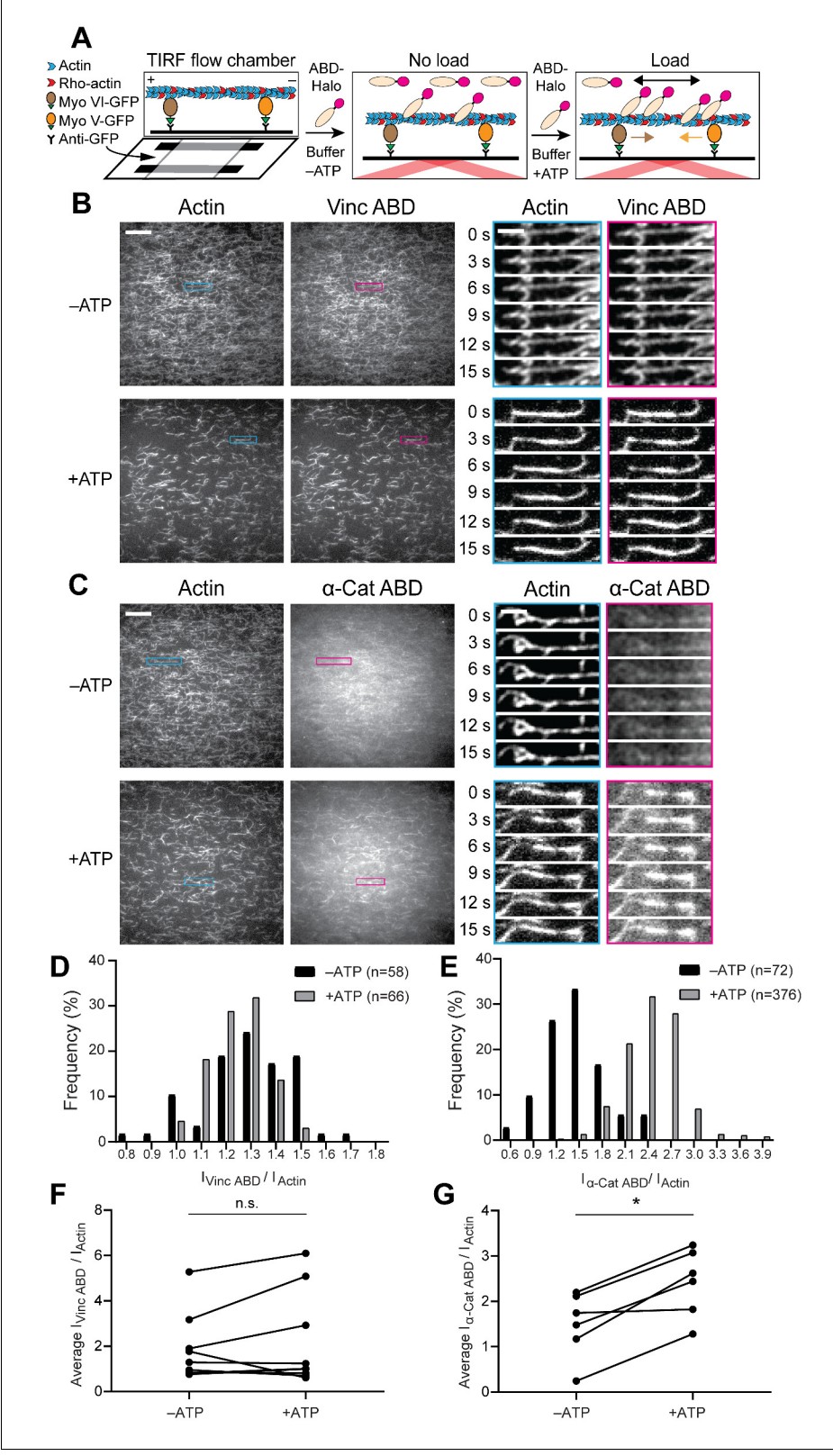

**Figure 2.** Forces generated by myosin motors activate α-catenin binding to F-actin. (**A**) Cartoon of TIRF force reconstitution assay utilizing surface-anchored myosin motor proteins with opposing directionality. (**B and C**) TIRF assays of vinculin ABD (**B**) and α-catenin ABD (**C**). Left: Representative movie frames of actin and ABD channels in the absence (top) and presence (bottom) of ATP to activate motors. Scale bar, 20 μm. Right: Montages of individual filament regions. Scale bar, 5 μm. Concentration of vinculin ABD or α-catenin ABD: 2 μM. (**D and E**) Time-averaged $I_{ABD}/I_{actin}$ intensity ratio

*Figure 2 continued on next page*

*Figure 2 continued*

distributions of filament regions before and after ATP addition for vinculin ABD (D, quantification of (B)) and α-catenin ABD (E, quantification of (C)). (F and G) Paired analysis of the overall average intensity ratio change before and after ATP addition for vinculin ABD (F) (N = 9, p=0.50) and α-catenin ABD (G) (N = 6, p=0.031). Wilcoxon signed-rank test: *p<0.05; n.s. (not significant), p>0.05.

The online version of this article includes the following video and figure supplement(s) for figure 2:

**Figure supplement 1.** Workflow of TIRF data quantification, additional intensity ratio distributions of single-filament regions for WT α-catenin and vinculin ABDs, and TIRF assays with the metavinculin ABD.

**Figure supplement 2.** Single motors do not activate actin binding by the vinculin or α-catenin ABD.

**Figure supplement 3.** Additional analysis of TIRF assays.

**Figure 2—video 1.** TIRF force reconstitution assay of wild-type vinculin ABD.

https://elifesciences.org/articles/62514#fig2video1

**Figure 2—video 2.** TIRF force reconstitution assay of wild-type α-catenin ABD.

https://elifesciences.org/articles/62514#fig2video2

**Figure 2—video 3.** TIRF force reconstitution assay (myosin V only) of wild-type vinculin ABD.

https://elifesciences.org/articles/62514#fig2video3

**Figure 2—video 4.** TIRF force reconstitution assay (myosin V only) of wild-type α-catenin ABD.

https://elifesciences.org/articles/62514#fig2video4

**Figure 2—video 5.** TIRF force reconstitution assay (myosin VI only) of wild-type vinculin ABD.

https://elifesciences.org/articles/62514#fig2video5

**Figure 2—video 6.** TIRF force reconstitution assay (myosin VI only) of wild-type α-catenin ABD.

https://elifesciences.org/articles/62514#fig2video6

As we expected the random deposition and inherent stochasticity of molecular motors in our assay to give rise to a distribution of forces and ABD binding states, we implemented an image analysis procedure to quantify ABD association by automatically identifying and tracking tens to hundreds of 'filament regions' through time (Materials and methods, *Figure 2—figure supplement 1A*). For each region, $I_{ABD}/I_{actin}$ was calculated in each frame, then averaged over all frames in which the region was detected. Consistent with our qualitative interpretation, histograms of $I_{ABD}/I_{actin}$ distributions before and after ATP addition to individual flow chambers showed no shift for the vinculin ABD (*Figure 2D*, *Figure 2—figure supplement 1B*). However, this analysis demonstrated a clear shift toward higher values upon ATP addition for the α-catenin ABD, supporting force-activated binding (*Figure 1E*, *Figure 2—figure supplement 1B*). While the reported trends were consistent across experiments for both ABDs, we nevertheless observed variability between trials (*Figure 2—figure supplement 1B*), potentially due to differences in the background intensities in both channels resulting from inconsistencies in cover-glass surface preparation (Materials and methods). We thus performed paired analysis of the mean $I_{ABD}/I_{actin}$ between the -ATP/+ATP conditions for each chamber (*Figure 2F and G*), which demonstrated a significant increase only for the α-catenin ABD (*Figure 2G*).

Our optical-trapping experiments suggest that force along individual filaments is sufficient to activate α-catenin binding. However, in the cellular context, both α-catenin and vinculin primarily engage actin-myosin bundles. In our TIRF assay, visual inspection supports increased F-actin bundling by both vinculin and α-catenin upon motor activation (*Figure 2B and C*), presumably due to motility facilitating encounters between filaments. Although $I_{ABD}/I_{actin}$ measurements are internally normalized for the local density of F-actin in each region, we are aware that inter-filament ABD bundling contacts could in principle enhance apparent binding. Additionally, while the ATP-dependence of this enhanced F-actin binding strongly suggests it is activated by force, allosteric remodeling of actin filament structure due to local deformations imposed by motor binding could also potentially contribute (*Gurel et al., 2017*). To decouple these effects, we performed assays in the presence of individual motors. In the presence of both myosin V alone (*Figure 2—figure supplement 2*, *Figure 2—video 3* and *Figure 2—video 4*) and myosin VI alone (*Figure 2—figure supplement 2*, *Figure 2—video 5* and *Figure 2—video 6*) we observe ATP-dependent formation of bundles. However, we observe no significant increase in apparent vinculin ABD or α-catenin ABD binding in either condition. This strongly suggests that force-activated

α-catenin ABD binding is dependent upon the tug-of-war between motors of opposed directionality, mimicking the forces generated by bi-polar myosin II filaments in vivo.

We note that forces generated by the randomly distributed force generators in the dual-motor assay are complex, and can in principle include tension, compression, and torsional forces (*Beausang et al., 2008*; *Sun et al., 2007*). While our optical trapping studies suggest tensile forces are sufficient to activate α-catenin ABD binding (*Figure 1*), future studies will be required to explicitly dissect the contribution of compression and torsion in the presence of myosin motors.

Additionally, we note that there are caveats associated with our TIRF experiments. First, the relatively low apparent affinity of our α-catenin ABD construct in this assay necessitates utilizing a high working concentration. The commensurate high fluorescence background is refractory to definitively measuring the intensity level constituting F-actin binding saturation in the absence or presence of force generation. This limits our ability to establish the binding stoichiometry range in which force-activated actin binding occurs, which has the potential to constrain or discriminate between plausible molecular mechanisms (see Discussion). In contemporary studies, Xu and colleagues have reported a series of α-catenin ABD N-terminal deletion constructs with substantially increased F-actin affinity in the absence of force, notably a construct consisting of residues 698–906, which boosts affinity ~18-fold (*Xu et al., 2020*). Presuming this construct maintains force-activated binding activity, it may be a useful tool to dissect the impact of α-catenin's binding stoichiometry. Second, due to the strong propensity of the α-catenin ABD to bundle actin filaments in our TIRF experiments, it remains to be conclusively determined if myosin motor activity can also activate α-catenin binding along single filaments. It may be possible to achieve a sufficiently low surface density of actin filaments to avoid bundling in the dual-motor assay in presence of the actin stabilizing drug phalloidin, which would enable probing α-catenin's association with individual actin filaments in the presence of active force generation. We will pursue these studies, and their outcome will be presented in a follow-up report.

Regardless, the data we present here collectively suggest physiological forces generated by myosin motor proteins in an appropriate configuration can directly activate α-catenin binding to F-actin, and that force-activated α-catenin binding also occurs in the context of actin bundles, the primary cytoskeletal architecture engaged by the protein in vivo.

## Cryo-EM structures of the metavinculin ABD-F-actin and α-catenin ABD-F-actin complexes

Hypothesizing that differences in the F-actin-binding interfaces of vinculin and α-catenin could underlie their differential force-activated actin binding, we pursued structural studies of both ABDs bound to F-actin with cryo-EM (*Figure 3*; *Figure 3—figure supplement 1*; *Table 1*). As optimizing the density of fully-decorated, well-separated individual filaments in cryo-EM images is a major bottleneck for single-particle analysis of F-actin-ABP complexes, we chose to use the ABD of the vinculin splice variant metavinculin for these studies, where a 68 amino-acid insert displaces the H1 helix and replaces it with helix H1', producing a protein which retains actin binding but completely loses actin bundling activity (*Janssen et al., 2012*; *Kim et al., 2016*; *Oztug Durer et al., 2015*; *Figure 1—figure supplement 1B*). Previous studies have suggested these isoforms engage an essentially identical site on the F-actin surface with equivalent affinity (*Janssen et al., 2012*; *Kim et al., 2016*), and we further found the metavinculin ABD lacks force-activated actin-binding activity in our TIRF assay, validating its use in these studies (*Figure 2—figure supplement 1C,D*). We were able to obtain fields of individual decorated filaments using this construct (*Figure 3A*, top). After careful optimization (Materials and methods), we were also able to acquire cryo-EM images of filaments decorated with the α-catenin ABD (*Figure 3A*, bottom), although persistent bundling by this construct necessitated the collection of substantially more images to obtain a sufficient dataset of individual segments to obtain a high-resolution reconstruction (*Table 1*; *Figure 3—figure supplement 1*).

Using the Iterative Helical Real Space Reconstruction (IHRSR) approach (*Egelman, 2007*) as implemented in Relion 3.0 (*He and Scheres, 2017*; *Zivanov et al., 2018*; Materials and methods, *Figure 3—figure supplement 1*), we obtained reconstructions of the metavinculin ABD (residues 879–1134)–F-actin complex (*Figure 3B*, left) at 2.9 Å overall resolution (*Figure 3—figure supplement 1*) and the α-catenin ABD (residues 664–906)–F-actin complex (*Figure 3B*, right) at 3.2 Å overall resolution (*Figure 3—figure supplement 1*). As local resolutions ranged from 2.7 Å to 3.6 Å,

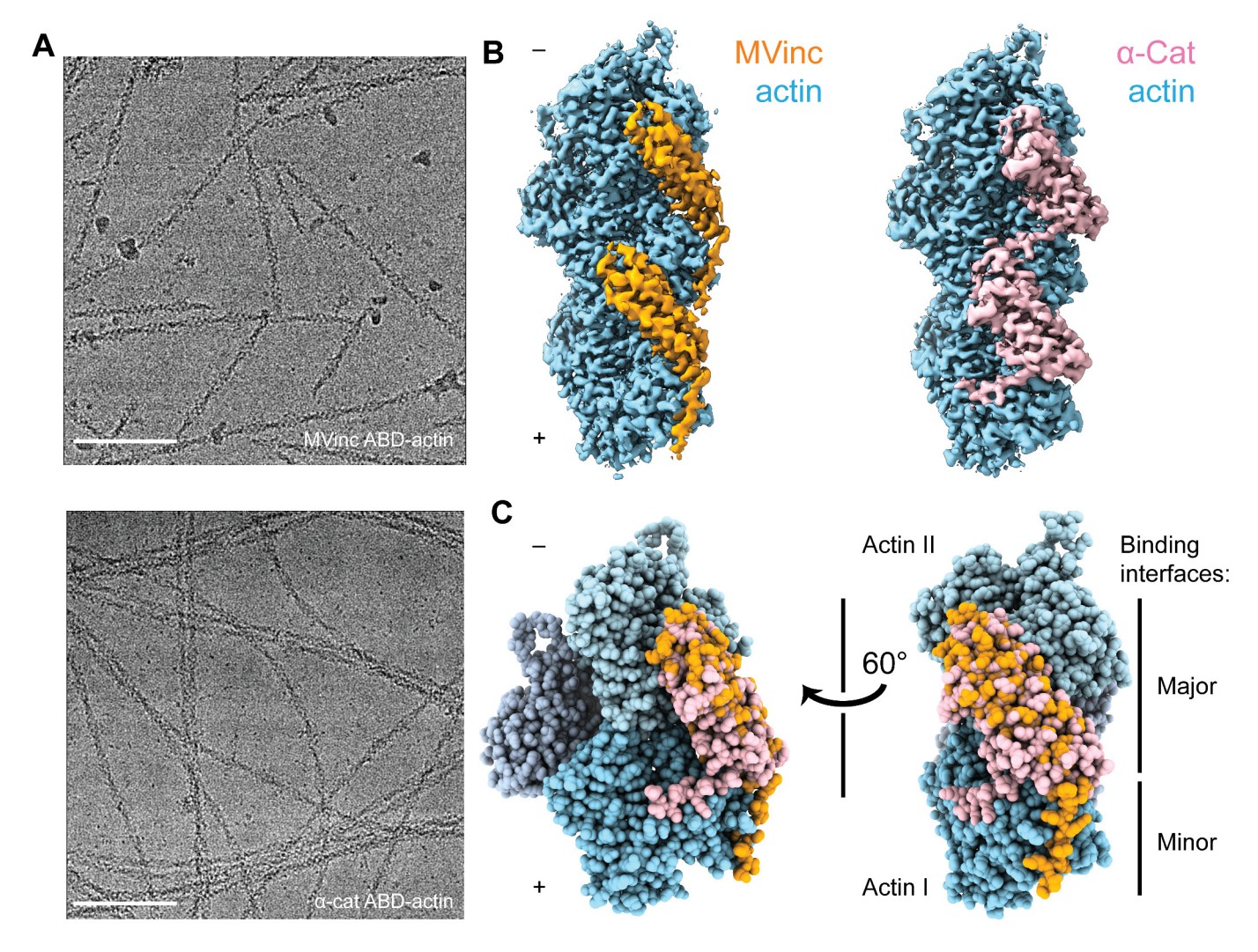

**Figure 3.** Cryo-EM structures of metavinculin and α-catenin–F-actin complexes. (**A**) Representative cryo-EM micrographs of F-actin decorated with metavinculin ABD (top) and α-catenin ABD (bottom). Scale bars, 100 nm. (**B**) Segmented regions of reconstructed density maps of F-actin decorated with metavinculin ABD (left) and α-catenin ABD (right), colored as indicated. (**C**) Overlay of the metavinculin ABD-actin complex and α-catenin ABD-actin complex atomic models, superimposed on actin and colored as in (**B**). Actin subunits from the α-catenin structure are displayed in varying shades. The online version of this article includes the following figure supplement(s) for figure 3:

**Figure supplement 1.** Cryo-EM data processing workflow and 3D reconstruction analysis.

radially decaying outward from the helical axis (*Figure 3—figure supplement 1C,D*) for both reconstructions, atomic models for the complete sequence of Mg-ADP α-actin and continuous segments of ABD residues 981–1131 for metavinculin (*Figure 3C*) and 699–871 for α-catenin (*Figure 3C*) were built and refined into the maps (*Figure 3—figure supplement 1*). In their contemporary work, Xu and colleagues also reported a 3.6 Å resolution cryo-EM reconstruction of the α-catenin ABD bound to F-actin (*Xu et al., 2020*), which shows a highly similar conformation of the complex to that presented here (actin, 0.5 Å RMSD; α-catenin residues 711–842, corresponding to helices H2–H5, 1.0 Å RMSD).

Superposition of the actin-bound metavinculin ABD with the full-length vinculin crystal structure (*Bakolitsa et al., 2004*) confirms that actin binding by both vinculin isoforms is auto-inhibited by intramolecular interactions between the N-terminal head and C-terminal ABD tail

**Table 1.** Cryo-EM data collection, refinement, and validation statistics.

| | Metavinculin ABD–F-actin (EMDB-20844, PDB 6UPW) | α-catenin ABD–F-actin (EMDB-20843, PDB 6UPV) |
|---|---|---|
| **Data collection and processing** | | |
| Microscope | Titan Krios | Titan Krios |
| Voltage (kV) | 300 | 300 |
| Detector | K2 Summit | K2 Summit |
| Magnification | 29,000 | 29,000 |
| Electron exposure (e⁻/Å$^2$) | 60 | 60 |
| Exposure rate (e⁻/pixel/ s) | 6 | 6 |
| Calibrated pixel size (Å) | 1.03 | 1.03 |
| Defocus range (μm) | –1.5 to –3.5 | –1.5 to –3.5 |
| Helical symmetry | C1 | C1 |
| | 27.06 Å rise | 27.03 Å rise |
| | −167.07° twist | −166.88° twist |
| Initial filament segments (no.) | 237,503 | 540,533 |
| Final filament segments (no.) | 215,369 | 414,486 |
| Map resolution (Å) | 2.94 | 3.24 |
| FSC threshold | 0.143 | 0.143 |
| **Refinement** | | |
| Initial model (PDB ID) | 3JBK, 3J8A | 4IGG, 3J8A |
| Model resolution (Å) | 2.98 | 3.28 |
| FSC threshold | 0.5 | 0.5 |
| Map sharpening B factor (Å$^2$) | –58 | –85 |
| Model composition | 5 actin protomers, 2 actin-binding domains | |
| Non-hydrogen atoms | 16,896 | 17,233 |
| Protein residues | 2152 | 2201 |
| Ligands | 5 Mg.ADP | |
| **B factors (Å$^2$)** | | |
| Protein | 53.41 | 71.59 |
| Ligand | 47.29 | 56.65 |
| R.M.S. deviations | | |
| Bond lengths (Å) | 0.012 | 0.012 |
| Bond angles (°) | 0.860 | 0.833 |
| **Validation** | | |
| MolProbity score | 1.23 | 1.33 |
| Clash score | 3.32 | 4.63 |
| Poor rotamers (%) | 0.06 | 0.05 |
| **Ramachandran plot** | | |
| Favored (%) | 97.46 | 97.56 |
| Allowed (%) | 2.54 | 2.44 |
| Disallowed (%) | 0.00 | 0.00 |
| EMRinger Score | 4.40 | 3.87 |

domains (*Johnson and Craig, 1995*; *Figure 4—figure supplement 1A,B*), as the head domain clearly clashes with F-actin in the crystallized conformation. Full-length α-catenin was crystallized as an asymmetric 'left-handshake dimer', characterized by differential relative orientations between the head and tail domains of each protomer (*Rangarajan and Izard, 2013*). Comparing the actin-bound α-catenin ABD with both conformers in the asymmetric dimer crystal structure also reveals severe clashes between the α-catenin head and actin (*Figure 4—figure supplement 1C–F*), suggesting that full-length α-catenin must also undergo substantial conformational rearrangements to bind F-actin.

Next, we compared the metavinculin ABD-F-actin and α-catenin ABD-F-actin structures (*Figure 3C*), confirming previous low- and moderate-resolution studies (*Janssen et al., 2006*; *Janssen et al., 2012*; *Kim et al., 2016*; *Thompson et al., 2014*) that both ABDs engage a major site spanning the longitudinal interface of 2 actin protomers, which we term Actin I and Actin II (numbered from the plus end of the filament). In turn, each actin protomer also contacts 2 ABDs, leading to a 1:1 binding stoichiometry at saturation. Our structures establish this region is almost identical between the two ABDs, comprising 2040 $\text{Å}^2$ of buried surface area for the metavinculin ABD and 1920 $\text{Å}^2$ for the α-catenin ABD. However, our high-resolution models also reveal a previously unobserved minor interface for each ABD (confirming a recent computational prediction in the case of metavinculin [*Krokhotin et al., 2019*]), mediated by residues in their flexible C-terminal extensions (CTEs), which are entirely distinct between the two proteins (*Figure 1—figure supplement 1A*).

## Differential conformational remodeling is linked to unique features of actin binding

Consistent with our previous medium-resolution structural studies (*Kim et al., 2016*), we find the metavinculin ABD undergoes substantial conformational remodeling upon F-actin engagement, characterized by displacement of helix H1' from the 5-helix bundle to license a rearrangement of helices H2–H5 to relieve clashes with F-actin (*Figure 4A,B*; *Figure 4—video 1* and *Figure 4—video 2*). N-terminal residues 879–980 are not visible in the map (*Figure 4A*, transparent brown), and are presumably disordered in the actin-bound state, while residues 981–985 (*Figure 4A*, brown) undergo a slight rearrangement, extending helix H2 by 1.5 turns (five residues). Our high-resolution map reveals this contact to be mediated by the (meta)vinculin CTE (*Figure 4*). The CTE is released from its pre-bound position, extending helix H5 by two turns (six residues), then undergoing an approximately 60° swing to engage a site along actin subdomain one proximal to H5 (*Figure 4A and B*; *Figure 5—video 1*). Coupled to this transition, helices H2–H5 slightly rearrange to accommodate actin binding and avoid steric clashes (*Figure 4B and C*; *Figure 4—video 2*).

We find that the α-catenin ABD also undergoes an order-to-disorder transition at its N-terminus upon actin binding (*Figure 4A,B*; *Figure 4—video 1*), as no density for residues 664–698, the majority of H0-H1, is present in our map (*Figure 3D*; *Figure 4A,B*; *Figure 4—video 1*), confirming recent reports that this region is important for activating α-catenin's actin engagement (*Ishiyama et al., 2018*; *Xu et al., 2020*). This is accompanied by a twisting rearrangement of helices H2–H4 reminiscent of that found in (meta)vinculin (*Figure 4B*; *Figure 4—video 2*), as well as the extension of H4 by three turns (nine residues) through folding of the H3-H4 loop, to sculpt a major actin-binding interface sterically compatible with the filament (*Figure 4B*; *Figure 4—video 1*). This suggests that N-terminal helix release allosterically coupled to ABD helical-bundle rearrangement is a fundamentally conserved mechanistic feature of actin binding by members of the vinculin/α-catenin family. However, we observe distinct rearrangements in the α-catenin CTE, which undergoes a slight lateral shift and helical unfurling, rather than a swing, to engage a site spanning a different surface of actin subdomain 1 (*Figure 4A and B*; *Figure 5—video 1*).

Metavinculin H5 extension is facilitated by binding interactions with Actin I (*Figure 5A,B*; *Figure 5—video 1*), notably a hydrophobic interaction between metavinculin I1114 and Actin Y91, and a salt bridge between metavinculin R1117 and actin E100. This positions the CTE to form an extended interface with actin subdomain 1, contiguous with that mediated by H5, with metavinculin W1126 buried in a proximal hydrophobic pocket formed by actin residues A7, P102, P130, A131, and W356, bolstered by a distal salt bridge between metavinculin R1128 and actin E361 (*Figure 5B*). By contrast, the α-catenin CTE retains an overall conformation similar to its pre-bound state (*Figure 5A and C*; *Figure 5—video 1*). An extensive hydrophobic network we term a

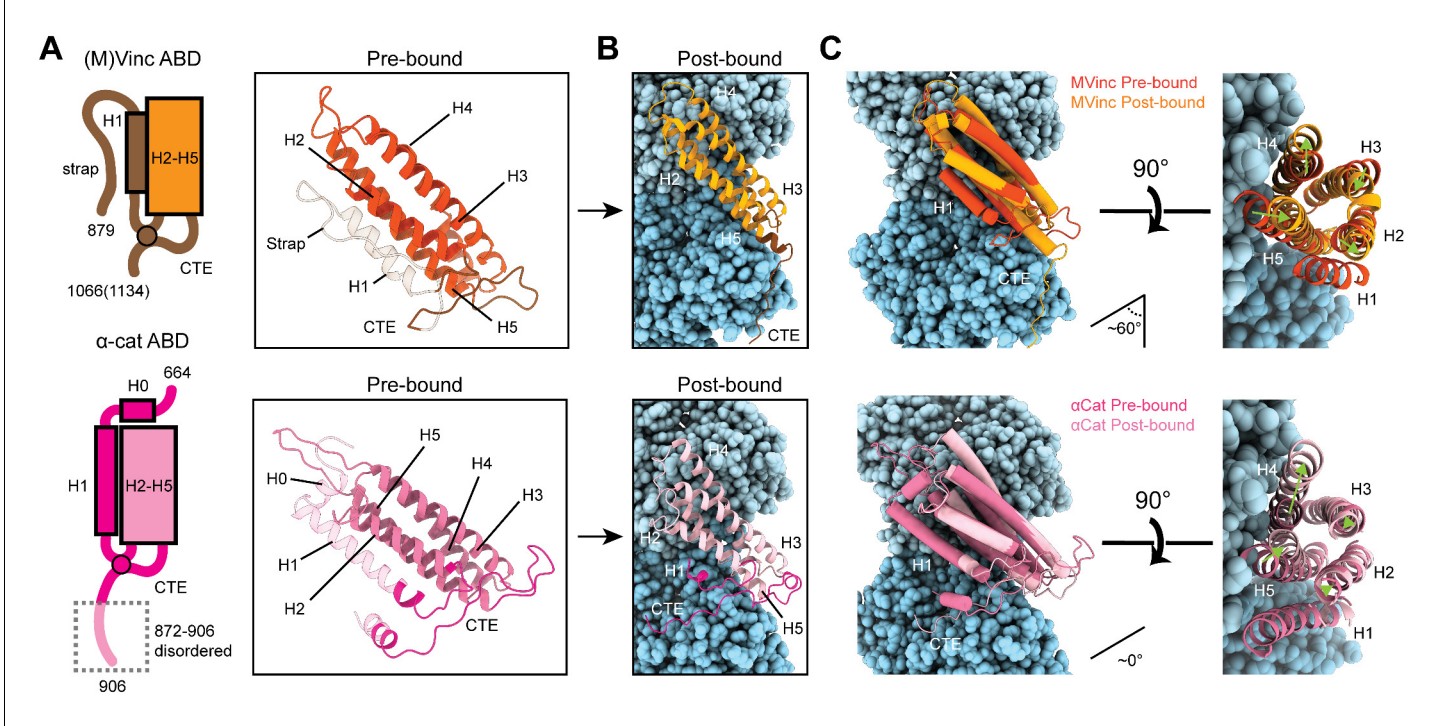

**Figure 4.** Major ABD conformational rearrangements upon actin binding. (**A**) Cartoons and crystal structures of pre-bound (meta)vinculin ABD (PDB 1st6, dark orange, top) and α-catenin ABD (PDB 4igg chain B, dark pink, bottom). Regions that undergo major rearrangements are highlighted in brown (metavinculin) or magenta (α-catenin); regions not resolved in the actin-bound cryo-EM structures are transparent. (**B**) Cryo-EM structures of metavinculin ABD (orange, top) and α-catenin ABD (pink, bottom). Flexible regions colored as in (**A**); actin, shades of blue. (**C**) Superimposed pre-bound and post-bound structures of metavinculin ABD (top) and α-catenin ABD (bottom). Green arrows indicate displacement of helices; rotation angles indicate repositioning of C-terminal extensions (CTEs).

The online version of this article includes the following video and figure supplement(s) for figure 4:

**Figure supplement 1.** (Meta)vinculin and α-catenin activation mechanisms.

**Figure 4—video 1.** Overall conformational changes in ABDs upon actin binding.

https://elifesciences.org/articles/62514#fig4video1

**Figure 4—video 2.** Rearrangements in the helical bundle regions of ABDs upon actin binding.

https://elifesciences.org/articles/62514#fig4video2

'tryptophan latch' embraces CTE residue W859 in both pre-bound and post-bound states, preventing α-catenin CTE unfurling (*Figure 5C*, right; *Figure 5—video 1*). A single turn of helix H1 on the N-terminal side of the ABD remains folded, with H1 residue W705 packing against CTE residue M861, encircling W859 along with CTE residues L852 and L854, as well as residues W705, I712, I763, L776, P768, V833, and Y837 from neighboring regions of the helical bundle, facilitating coordinated conformational transitions between the N- and C-terminal flexible regions of the α-catenin ABD upon actin binding. A putative hydrogen bond is also maintained between S840 and the single nitrogen atom in W859's indole ring, maximizing the binding potential of this residue.

The latch positions the neighboring region of the CTE to bind a distinct site on Actin I's subdomain 1 (*Figure 5C*, left; *Figure 5—video 1*) mediated by proximal salt bridges (α-catenin E865 – actin R28, α-catenin K866 – actin D24/D25, α-catenin K867 – actin E93) and distal hydrophobic interactions (α-catenin L869/V870 – actin P333/Y337). Consistent with a key role for the latch in coordinating conformational transitions that enable F-actin binding, Xu et al. report that mutating W859 to alanine reduces α-catenin's F-actin-binding affinity 10-fold (*Xu et al., 2020*). Superposition of the actin-bound conformation of the α-catenin ABD with the pre-bound conformation of the (meta)vinculin ABD (*Figure 4—figure supplement 1G*) reveals a striking positional overlap between α-catenin W859 and metavinculin W1126 (vinculin W1058 is identically positioned, not shown), which is also

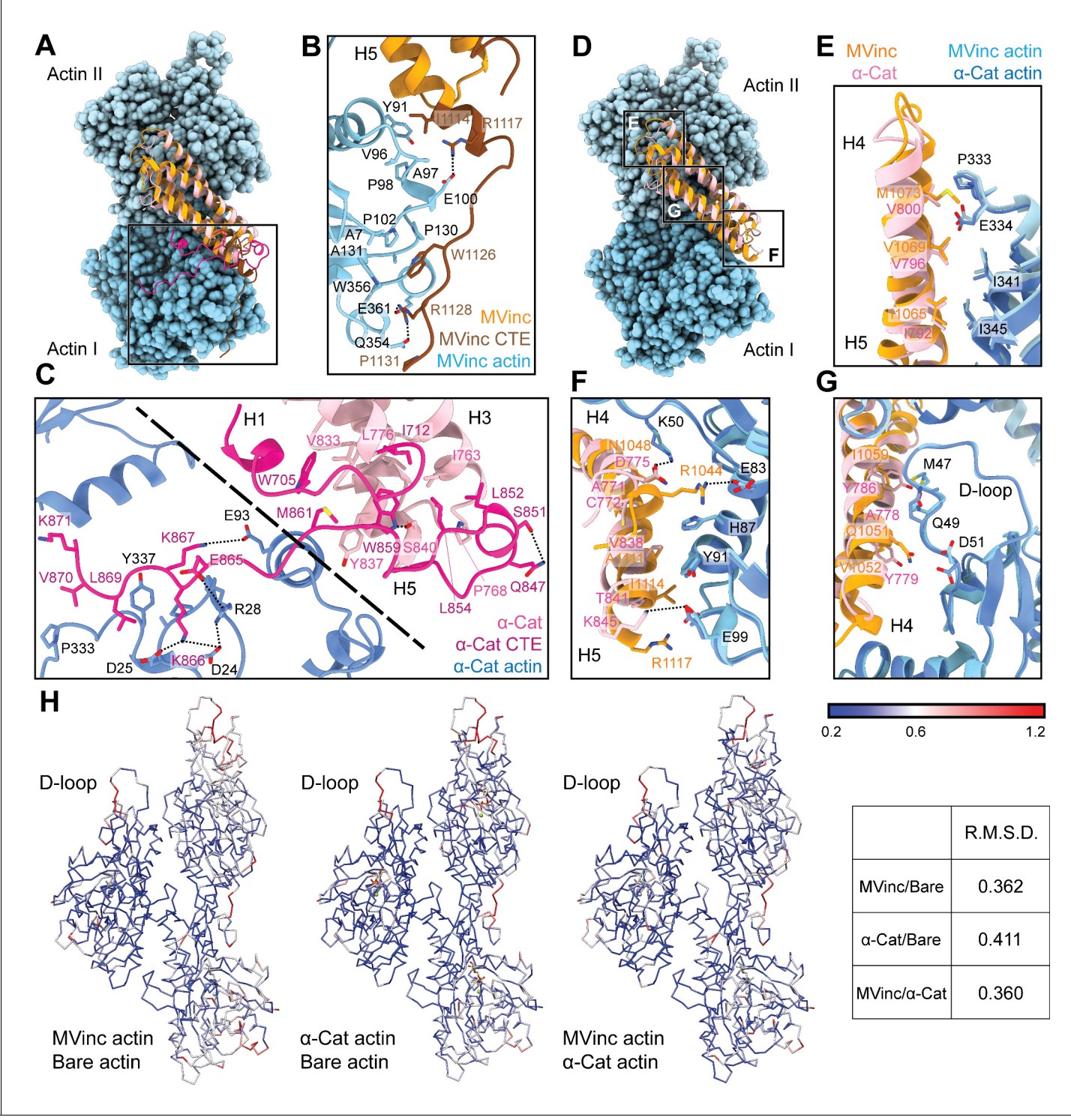

**Figure 5.** The actin-binding interfaces of metavinculin and α-catenin. (**A**) Overlay of the metavinculin ABD-actin complex and α-catenin ABD-actin complex atomic models highlighting C-terminal extensions (CTEs) of metavinculin ABD and α-catenin ABD, superimposed on Actin I and colored as in *Figure 4*. Actins from α-catenin structure are displayed. (**B and C**) Detailed views of key contacts at minor interfaces: metavinculin CTE and actin (**B**); within the α-catenin tryptophan latch (right) and between its CTE and actin (left) (**C**). (**D**) Overlay of the metavinculin ABD-actin complex and α-catenin ABD-actin complex atomic models highlighting helical binding interfaces of metavinculin ABD and α-catenin ABD, superimposed and colored as in (**A**). Actins from α-catenin structure are displayed. (**E, F, and G**) Detailed views of key contacts at the major interface between metavinculin/α-catenin helices H4–H5 and Actin I (**E**); Actin II (**F**); Actin II D-loop (**G**). (**H**) Actin C$_\alpha$ traces colored by per-residue RMSD from the indicated comparisons. For superposition, segmented actin density from the cryo-EM maps was first aligned, followed by fitting the atomic models into their corresponding maps.

*Figure 5 continued on next page*

*Figure 5 continued*

The online version of this article includes the following video and figure supplement(s) for figure 5:

**Figure supplement 1.** TIRF assays for the α-catenin ABD triple mutant which does not bind actin, and putative ABD-ABD contacts along F-actin.
**Figure 5—video 1.** Detailed views of distinct conformational changes in CTEs.
https://elifesciences.org/articles/62514#fig5video1

___

engaged by a sparser latch in the pre-bound conformation (*Figure 5—video 1*). We thus speculate the extensive latch of α-catenin prevents W859 release and the extension of its CTE to engage the same site as metavinculin W1126.

## Major filament binding interfaces discriminate actin D-loop conformations

The complete non-overlap of the (meta)vinculin and α-catenin minor actin-binding interfaces mediated by their CTEs lead us to hypothesize that the CTEs could be involved in differential force-activated actin binding. To identify other potential contributing structural elements, we undertook a detailed comparison of their major actin-binding interfaces mediated by helices H4–H5 in both proteins (*Figure 5D*). The Actin II binding interface is almost identical between the two ABDs (*Figure 5E*), mediated by an extensive series of conserved hydrophobic contacts: α-catenin I792/metavinculin I1065 – actin I345, α-catenin V796/metavinculin V1069 – actin I341, and α-catenin V800/metavinculin M1073 – actin P333/E334. The Actin I interface, on the other hand, is more variable and characterized by few clear residue-level binding interactions (*Figure 5F*), notably likely weak long-distance salt bridges (metavinculin R1044 – actin E83 and α-catenin K845 – actin E99) specific to each protein, despite the overall shape complementarity across the interface. Each ABD also features a unique hydrophobic interaction with actin Y91 (metavinculin I1114/α-catenin V838).

The Actin I interface extends into contacts with the actin D-loop (*Figure 5G*), a flexible region of actin which mediates structurally polymorphic longitudinal interactions between protomers (*Galkin et al., 2010b*) reported to be modulated by actin nucleotide state (*Chou and Pollard, 2019*; *Merino et al., 2018*) and ABPs (*Dominguez and Holmes, 2011*; *Oda et al., 2019*). Both ABDs form a potential weak long-distance interaction with actin D-loop residue K50: in metavinculin, a hydrogen bond through N1048, and in α-catenin, a salt bridge through D775. The D-loop then adopts subtly different conformations between the two interfaces centered at M47. Although clear residue-level binding interactions are not readily apparent, the conformation of M47 at the metavinculin interface would clash with α-catenin Y786, a position occupied by the smaller residue I1059 in metavinculin, suggesting local sterics unique to each ABD determine compatibility with a distinct D-loop conformation. Comparison of the actin conformation observed in a similar-resolution structure of ADP F-actin in isolation ('F-actin alone', M.S. in preparation) versus when bound to metavinculin or α-catenin, as well as comparison of the metavinculin-bound and α-catenin-bound conformations reveals minimal rearrangements throughout the majority of the structure (*Figure 5H*). This contrasts with a previous report of α-catenin-induced structural changes in F-actin inferred from low-resolution cryo-EM analysis (*Hansen et al., 2013*), but it is consistent with the high-resolution studies of *Xu et al., 2020*. The sole region featuring rearrangements greater than 1 Å RMSD is a 3–4 residue stretch of the D-loop centered on M47. As force across the filament could feasibly modulate D-loop structure to regulate ABP binding, we hypothesized ABD residues mediating D-loop interactions could also mediate differential force-activated actin binding.

## The α-catenin C-terminal extension is a force-detector element

To investigate whether D-loop interactions contribute to α-catenin force-activated binding, we designed a triple point-mutant α-catenin ABD construct where three residues in close proximity to the D-loop were replaced by those in vinculin: α-cat ABD$^{A778Q, Y779V, Y786I}$. In force reconstitution assays, this construct did not visibly associate with actin in either the –ATP or +ATP condition in the concentration regime accessible by TIRF (*Figure 5—figure supplement 1A*). While these data

suggest that the α-catenin D-loop interactions contribute to overall affinity for F-actin, the complete lack of binding is refractory to determining whether this interface has a separable role in force-activated actin binding.

We thus returned to our initial hypothesis that differential force-activated binding could be mediated by the CTEs. Although we were unable to accurately model the final three residues of the metavinculin CTE, weak density is clearly present (*Figure 5—figure supplement 1B*, red), suggesting the entire CTE engages F-actin. By contrast, density for the α-catenin CTE is only present until K871 (*Figure 5C*; *Figure 5—figure supplement 1C*). Notably, all three human α-catenin isoforms have highly homologous CTEs that extend an additional 35 amino acids (*Figure 1—figure supplement 1A*), diverging in sequence and length from the vinculin CTE. Consistent with previous primary-structure-function analysis (*Pokutta et al., 2002*) showing that residues after P864, which bear no homology to vinculin, are necessary for actin binding, our structure shows that residues 865–871 are in direct contact with actin (*Figure 5C*; *Figure 5—video 1*), forming an extensive interface. We therefore hypothesized that distal residues 872–906, a 35-residue element unique to α-catenin (*Figure 1—figure supplement 1A*) that was not resolved in our cryo-EM analysis and is thus presumably conformationally flexible, could uniquely contribute to force-activated actin binding.

To test whether distal residues 872–906 have a separable role in force-activated binding, that is as a 'force detector', we first truncated them from the α-catenin ABD (α-cat ABDΔC). Consistent with a regulatory role for this segment, α-cat ABDΔC constitutively associated with F-actin in TIRF assays (*Figure 6A*; *Figure 6—figure supplement 1A*; *Figure 6—video 1*), with no significant increase in binding upon ATP addition, suggesting that this region is necessary for force-activated binding by negatively-regulating low-force binding. This contrasts with the observations of Xu et al., who report a modest (~2-fold) reduction in F-actin-binding affinity when this region is truncated in solution co-sedimentation assays, notably in the background of a construct where H0 has also been truncated in order to boost affinity overall (*Xu et al., 2020*). Possible sources of this discrepancy include differences in α-catenin binding behavior between solution assays and our TIRF assays, where filaments are immobilized. Additionally, coordination between rearrangements in the CTE and H0–H1 (potentially through allosteric mechanisms) may be necessary to mediate the force-detector's negative regulatory effects.

To investigate the sufficiency of the force-detector, we generated chimeric ABDs featuring the H2–H5 bundle region of vinculin and the flexible termini of α-catenin. A vinculin ABD construct where only the CTE was substituted was non-functional (data not shown). However, consistent with structural coordination between the α-catenin N-terminal segment and the CTE through the tryptophan latch (*Figure 5C*; *Figure 5—video 1*), a construct featuring both the α-catenin N-terminal segment and the CTE (vinc ABD-NCSwap, Materials and methods) gained force-activated binding activity, with diminished low-force binding observed (*Figure 6B*; *Figure 6—figure supplement 1B*; *Figure 6—video 2*) in contrast to the wild-type vinculin ABD (*Figure 1B*). A C-terminal truncation of this construct (vinc ABD-NCSwapΔC) equivalent to α-cat ABDΔC reverted to constitutive binding regardless of force (*Figure 6C*; *Figure 6—figure supplement 1C*; *Figure 6—video 3*), supporting the α-catenin CTE as the key determinant of force-activated binding, in which 872–906 serves as the force-detector. We thus conclude that the distal C-terminus (residues 872–906) of α-catenin is necessary and sufficient for force-activated actin binding through negative regulation of low-force binding.

## Discussion

Our studies reveal that α-catenin preferentially binds tensed F-actin, using its flexible C-terminus to detect piconewton load that can be generated by myosin motors. We propose this force-activated F-actin binding could play an important role in the formation and reinforcement of cell-cell adhesion complexes, facilitating mechanically regulated interactions between α-catenin and actomyosin cables tuned for high sensitivity to motor-generated forces.

We further provide evidence that actin filaments are themselves tension sensors that can transduce mechanical force into regulated partner binding. Our finding that the force-detector element of α-catenin can be functionally transplanted to vinculin demonstrates that force-activated F-actin binding is a modular activity that can be conferred by short sequence motifs. We thus speculate

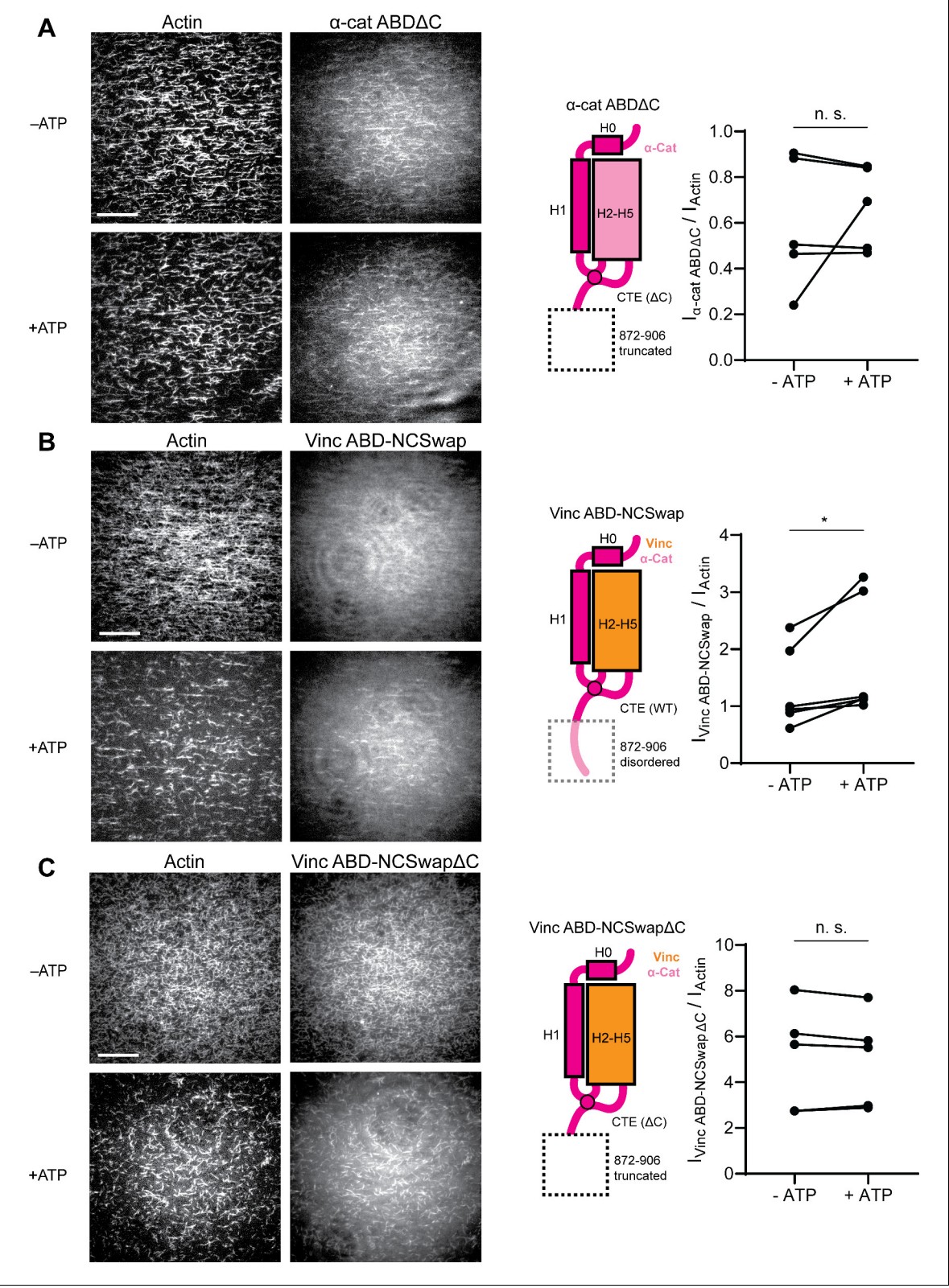

**Figure 6.** The distal tip of α-catenin's C-terminus is a force detector. (**A, B, and C**) TIRF force reconstitution assays. Left: Representative movie frames in the presence and absence of ATP. Scale bar, 30 μm. Right: Cartoon of ABD construct (left) and paired analysis of the overall intensity ratio change upon ATP addition (right). Wilcoxon signed-rank test: *p<0.05; n.s. (not significant), p>0.05. Constructs assayed were: α-cat ABDΔC (**A**) (αE-catenin$_{664-871}$,

*Figure 6 continued on next page*

*Figure 6 continued*

N = 5, p=0.81); Vinc ABD-NCSwap (**B**) ($\alpha$E-catenin$_{664-708}$-vinculin$_{916-1041}$-$\alpha$E-catenin$_{837-906}$, N = 6, p=0.031); Vinc ABD-NCSwap$\Delta$C (**C**) ($\alpha$E-catenin$_{664-708}$-vinculin$_{916-1041}$-$\alpha$E-catenin$_{837-871}$, N = 5, p=0.63). Concentration of ABD constructs: 2 $\mu$M.

The online version of this article includes the following video and figure supplement(s) for figure 6:

**Figure supplement 1.** Intensity ratio distributions of single-filament regions for $\alpha$-catenin and vinculin CTE mutants.

**Figure 6—video 1.** TIRF force reconstitution assay of $\alpha$-catenin ABD$\Delta$C truncation mutant.

https://elifesciences.org/articles/62514#fig6video1

**Figure 6—video 2.** TIRF force reconstitution assay of Vinc ABD-NCSwap chimera.

https://elifesciences.org/articles/62514#fig6video2

**Figure 6—video 3.** TIRF force reconstitution assay of Vinc ABD-NCSwap$\Delta$C chimera truncation mutant.

https://elifesciences.org/articles/62514#fig6video3

there may be other actin-binding proteins with functionally analogous force-detectors, which could broadly mediate mechanotransduction through the cytoskeleton.

## Molecular mechanism for detecting force on actin through a flexible structural element

While our studies pinpoint the final 35 amino acids of $\alpha$-catenin as a force-detector, the exact molecular mechanism by which this segment negatively regulates low-force binding to F-actin remains to be elucidated. Here we propose two potential, non-exclusive conceptual models for this modulation. As we observe the distal tip of the ordered region of the CTE to be in close apposition to the next ABD binding site along the actin filament, with potential contacts between CTE residues V870 and K871 with the H4–H5 loop and the N-terminal tip of H5 in longitudinally adjacent ABD (*Figure 5—figure supplement 1C*), the first model invokes steric exclusion (*Figure 7A*). In the absence of force, the force-detector (*Figure 6*; *Figure 7*) physically blocks the adjacent binding site through steric hindrance, which can be relieved by an increase in protomer axial spacing in the presence of tension, consistent with prior truncation studies suggesting residues 884–906 may inhibit $\alpha$-catenin's actin binding (*Pappas and Rimm, 2006*). As the force-detector likely represents a flexibly tethered conformational ensemble, in the presence of thermal fluctuations we envision this would manifest as a tension-dependent increase of the binding on-rate at the site due to its increased fractional availability. Although saturating the filament for structural studies could lead to non-physiological inter-ABD interactions, cooperative F-actin binding by the $\alpha$-catenin ABD has previously been reported under non-saturating conditions (*Hansen et al., 2013*), and supplemental soluble ABD enhanced catch-bonding by the cadherin complex (*Buckley et al., 2014*), suggesting communication between actin-bound ABDs is likely to be physiologically relevant. We note that cooperative and inhibitory inter-ABD communication are not a priori mutually exclusive, and the interplay of these opposing effects could produce differential outcomes as a function of ABD concentration and filament load, an important subject for future studies.

The second model invokes a conformational change in the actin protomer that specifically occurs in the presence of mechanical load, which is recognized and preferentially bound by the force-detector, relieving inhibition (*Figure 7B*). Although our studies suggest only minor actin conformational changes when the binding is driven by mass action (*Figure 5H*), they do not rule out as yet unobserved actin conformations specifically evoked by force. Furthermore, while for simplicity we have framed both models in terms of discrete transitions between structural states, low piconewton forces could also modulate the intrinsic structural fluctuations of F-actin to control $\alpha$-catenin engagement through either mechanism, as has previously been speculated for cofilin (*Galkin et al., 2012*; *Hayakawa et al., 2011*; *Wioland et al., 2019*). Although currently technically prohibitive, structural studies of the $\alpha$-catenin ABD–F-actin interface in the presence of active force generation, as well as supporting functional experiments, will be necessary to dissect the interplay of these models.

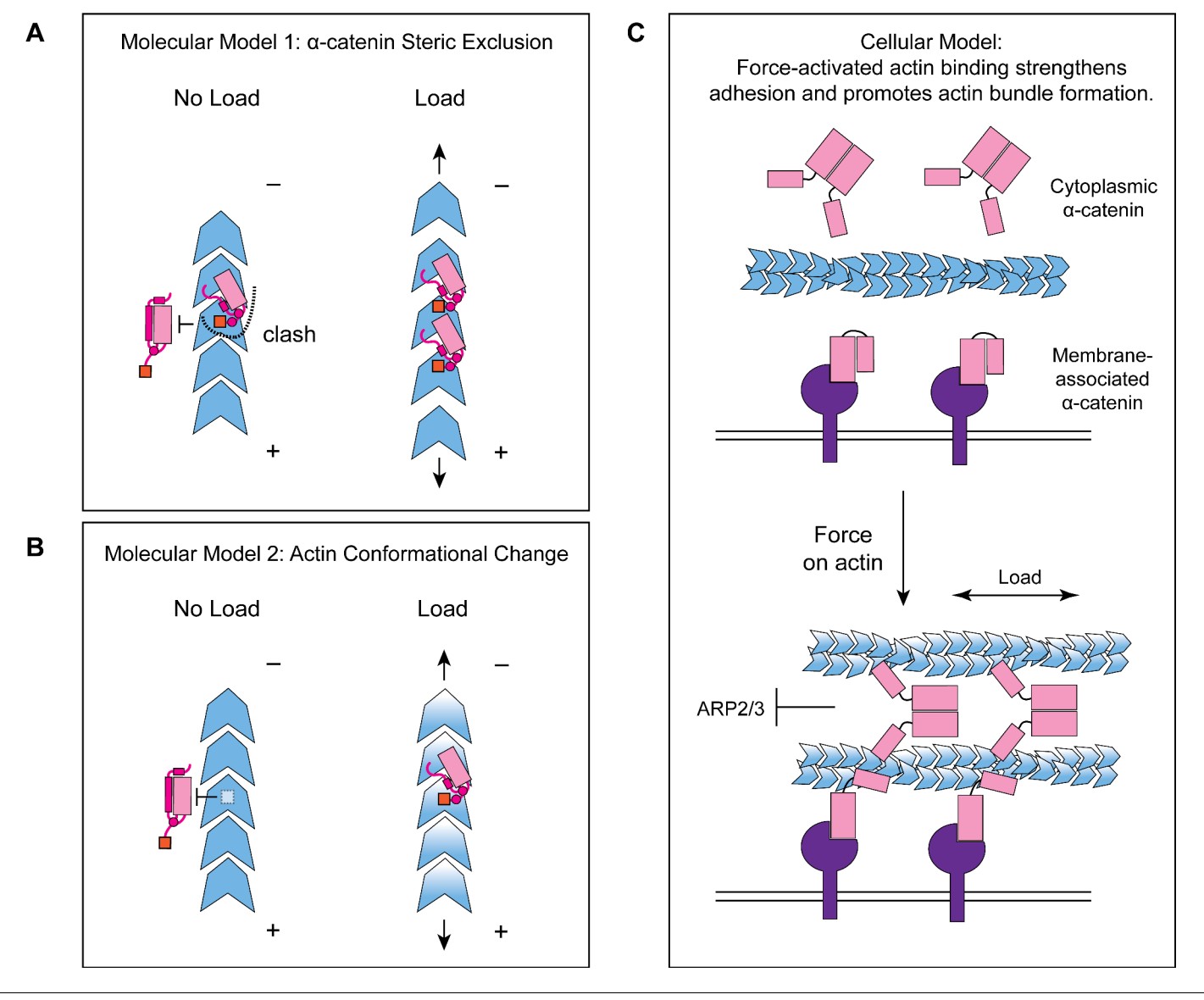

**Figure 7.** Molecular and cellular mechanisms for force-activated binding by α-catenin. (**A and B**) Cartoons of two conceptual mechanistic models for force-activated actin binding by the α-catenin ABD. Orange box represents the force detector (residues 872–906) at the distal tip of the CTE. Blue gradient represents a force-induced conformational transition in the actin protomer. (**C**) Cartoon of proposed biological functions for force-activated actin binding by the two cellular populations of α-catenin. Membrane-associated cadherin-catenin complexes are illustrated in purple.

## Mechanically-regulated adhesion through α-catenin and vinculin

Our finding that approximately 1 pN of tension along individual filaments is sufficient for force-activated binding suggests that the actin-binding interface of α-catenin has been evolutionarily optimized to sense contractile forces generated by myosin motors (*Finer et al., 1994*). We speculate force-activated binding enables the cadherin-catenin complex to recognize and preferentially engage pre-stressed actomyosin cables adjacent to the plasma membrane at adherens junctions (*Figure 7C*), providing a mechanism for initial engagement between actin and the cadherin-associated population of α-catenin, as well as rapidly strengthening adhesion after preliminary attachments are formed. This could facilitate the transition from nascent cell-cell contacts to mature adherens junctions, as punctate nascent adhesions associated with radial actin cables

coalesce and spread along a developing circumferential band of tensed actomyosin bundles (*Vaezi et al., 2002*), as well as support dynamic adherens junction remodeling during epithelial morphodynamics (*Lecuit and Yap, 2015*). It also provides a mechanism for concentrating the soluble, cytoplasmic α-catenin population at sites of cytoskeletal tension (*Figure 7C*). Previous studies have demonstrated this population is enriched on actomyosin bundles linked to adherens junctions, where it suppresses lamellipodium activity and promotes adhesion maturation (*Drees et al., 2005*). While enrichment has been speculated to occur via local release from the cadherin complex, this model is difficult to reconcile with its low cellular concentration and the high concentration of soluble α-catenin required to inhibit Arp2/3-mediated actin branching in vitro (*Drees et al., 2005*). Force-activated binding provides a feasible cellular mechanism for this enrichment of the homodimeric cytoplasmic α-catenin population, driven by enhanced affinity for tensed F-actin rather than mass action effects due to local concentration. Our identification of a functionally separable force-detector in α-catenin with amino-acid level precision will facilitate a detailed examination of these models in cell lines and in vivo.

We note that force-activated binding is an additional, rather than alternative, mechanism to catch-bond formation (*Buckley et al., 2014*) for mechanical regulation of F-actin binding. Our finding that vinculin lacks this activity despite forming catch-bonds with F-actin (*Huang et al., 2017*) strongly suggests that these two modes of mechanical regulation operate by unique structural mechanisms, likely to fulfill distinct biological functions. Vinculin's lack of force-activated actin-binding activity is consistent with the ordered sequence of mechanically-regulated binding events underlying its coordination with α-catenin at adherens junctions, where only after preliminary attachments form through the cadherin complex and come under load is vinculin recruited and activated to bind F-actin (*Yonemura et al., 2010*). The lower force threshold for force-activated binding than catch-bond formation (~1 pN vs. ~10 pN) supports a model in which the formation of initial attachments to the cytoskeleton through the cadherin complex is stimulated through force-activated binding, which is subsequently strengthened by catch-bonding through both α-catenin and vinculin during adhesion maturation.

In their parallel work, Xu et al. speculate that the displacement of H1 from the α-catenin ABD upon actin binding could be stabilized or induced by force (*Xu et al., 2020*), providing a possible structural mechanism for catch-bond formation which we have previously proposed could also be employed by vinculin (*Kim et al., 2016*; *Swaminathan et al., 2017*). This model predicts that the force required to dissociate H1 from the α-catenin ABD should be lower than the force required to displace the ABD in the mechanically-reinforced, strongly bound state from F-actin. It furthermore predicts that the constitutive high F-actin affinity α-catenin ABD construct (residues 711–842) lacking H0-H1 reported by Xu et al., which they propose mimics the strongly-bound state (*Xu et al., 2020*), should no longer form catch bonds with F-actin due to its anticipated inability to switch between weakly and strongly bound states. Thus, this proposed structural mechanistic framework is now well-positioned to be subjected to explicit experimental scrutiny. We believe dissecting the interplay between force-activated binding and catch-bond formation in vitro and in vivo, as well as the structural basis for coordinated actin catch-bonding by α-catenin and vinculin, are important subjects for future studies.

## Implications for cytoskeletal mechanotransduction

Our direct observation of force-activated binding to F-actin mediated by a short, flexible sequence element suggests this mechanism could feasibly be employed by other ABPs. Proteins from the large Calponin Homology (CH) domain ABP superfamily have diverse functions as cytoskeletal cross-linkers and plasma membrane/organelle tethers (*Liem, 2016*; *Razinia et al., 2012*). Their ABDs have also been reported to have sequence elements that undergo folding transitions associated with filament engagement (*Avery et al., 2017*; *Galkin et al., 2010a*; *Iwamoto et al., 2018*) which can sterically regulate their actin binding (*Harris et al., 2019*). This suggests members of this family could plausibly employ force-activated binding mechanisms similar to α-catenin to coordinate diverse mechanotransduction pathways throughout the cell, as could other ABPs with similar properties. The experimental strategy established here should be broadly useful for identifying force-sensitive ABPs and defining their force-detector-F-actin interfaces, such as the α-catenin CTE-F-actin interaction, in atomistic detail. This will facilitate elucidating the

molecular and cellular mechanisms of cytoskeletal mechanotransduction with sufficient precision for guiding inhibitor development.

# Materials and methods

## Key resources table

| Reagent type (species) or resource | Designation | Source or reference | Identifiers | Additional information |
|---|---|---|---|---|
| Recombinant DNA Reagent | pET His-Strep-TEV and pET His-Strep-TEV-Halo vectors | This paper | | Ampicillin resistance; expression vectors for *E. coli* Contact Alushin Lab for distribution |
| Recombinant DNA Reagent | pFastBac Dual Expression Vector | ThermoFisher | 10712024 | Expression vectors for sf9 insect cells |
| Gene (*H. sapiens*) | Vinculin; *Hs*VCL isoform 1 | Synthesized by Genewiz | UniProt:P18206-2 | |
| Gene (*H. sapiens*) | Metavinculin; *Hs*VCL isoform 2 | Synthesized by Genewiz | UniProt:P18206-1 | |
| Gene (*H. sapiens*) | αE-catenin; *Hs*CTNNA1 | Addgene # 24194 | UniProt:P35221 | |
| Strain, strain background (*E. coli*) | Rosetta2 (DE3) | Novagen | 71400–4 | *E. coli* strain for protein expression |
| Cell line (*S. frugiperda*) | Gibco Sf9 insect cell | ThermoFisher | 11496015 | Insect cell line for protein expression |
| Transfected construct (*S. frugiperda*) | Gibco Bac-to-Bac Baculovirus Expression System | ThermoFisher | 10359016 | Insect cell line transfection kit for protein expression |
| Chemical compound, drug | Halo-tag ligand amine O4 | Promega | P6741 | Synthetic precursor for the fluorescent dye JF-646-Halo |
| Chemical compound, drug | JF-646 NHS-ester building block | TOCRIS | 6148 | Synthetic precursor for the fluorescent dye JF-646-Halo |
| Chemical compound, drug | HaloTag Alexa Fluor 488 Ligand | Promega | G1001 | |
| Chemical compound, drug | Alexa Fluor 555-Phalloidin | ThermoFisher | A34055 | |
| Biological sample (*O. cuniculus*) | Actin Protein (Rhodamine): Rabbit Skeletal Muscle | Cytoskeleton, Inc | AR05-C | |
| Biological sample (*O. cuniculus*) | Actin Protein (Biotin): Rabbit Skeletal Muscle | Cytoskeleton, Inc | AB07-C | |
| Other | Streptavidin Coated Polystyrene Particles | Spherotech | SVP-40–5 | Used in optical trapping experiments |
| Other | C-flat Holey Carbon Grids for TEM - Gold Only | EMS | CF313-100-Au | Used in cryo-EM experiments |
| Software, algorithm | C-trap viewer | LUMICKS | | http://www.nat.vu.nl/~iheller/download.html |
| Software, algorithm | TWOM viewer | LUMICKS | | http://www.nat.vu.nl/~iheller/download.html |
| Software, algorithm | ImageJ 1.5 | *Schneider et al., 2012* https://doi.org/10.1038/nmeth.2089 | RRID:SCR_003070 | ImageJ plugin for TIRF image analysis: https://github.com/alushinlab/ActinEnrichment |

*Continued on next page*

*Continued*

| Reagent type (species) or resource | Designation | Source or reference | Identifiers | Additional information |
|---|---|---|---|---|
| Software, algorithm | RELION 3.0 | *He and Scheres, 2017* https://doi.org/10.1016/j.jsb.2017.02.003 *Schneider et al., 2012* https://doi.org/10.1016/j.jmb.2011.11.010 *Zivanov et al., 2018* https://doi.org/10.7554/eLife.42166 | RRID:SCR_016274 | |
| Software, algorithm | CTFFIND 4.1.5 | *Rohou and Grigorieff, 2015* https://doi.org/10.1016/j.jsb.2015.08.008 | RRID:SCR_016732 | |
| Software, algorithm | Phenix 1.17.1 | *Afonine et al., 2018* https://doi.org/10.1107/S2059798318006551 | RRID:SCR_014224 | |
| Software, algorithm | Coot 0.8.9 | *Emsley et al., 2010* https://doi.org/10.1107/S0907444910007493 | RRID:SCR_014222 | |
| Software, algorithm | UCSF Chimera 1.14 | *Pettersen et al., 2004* https://doi.org/10.1002/jcc.20084 | RRID:SCR_004097 | |
| Software, algorithm | UCSF ChimeraX 1.0 | *Goddard et al., 2018* https://doi.org/10.1002/pro.3235 | RRID:SCR_015872 | |
| Software, algorithm | GraphPad Prism 8 | GraphPad Software, Inc | RRID:SCR_002798 | |

## Contact and materials availability

Further information and requests for resources and reagents should be directed to the corresponding author, Gregory M. Alushin (galushin@rockefeller.edu). All reagents generated in this study are available from the corresponding author without restriction.

## Experimental model and subject details

Globular actin (G-actin) monomers were purified from chicken skeletal muscle as described previously (*Pardee and Spudich, 1982*) and maintained in G-Ca buffer: G buffer (2 mM Tris-Cl pH 8, 0.5 mM DTT, 0.2M ATP, 0.01% NaN$_3$) supplemented with 0.1 mM CaCl$_2$, at 4°C before use. C-terminally GFP-tagged mouse myosin V HMM and myosin VI S1 were purified from SF9 insect cells using published protocols (*Wang et al., 2000*). All other proteins were heterologously expressed in Rosetta2(DE3) *E. coli* cells (Novagen) grown in LB media as described in Method details.

## Method details

### F-actin preparation

Filamentous actin (F-actin) was polymerized fresh for each experiment from actin monomers in G-Mg (G buffer supplemented with 0.1 mM MgCl$_2$) and KMEI (50 mM KCl, 1 mM MgCl$_2$, 1 mM EGTA, 10 mM imidazole pH 7.0, 1 mM DTT) buffers as described previously (*Gurel et al., 2017*); 30% rhodamine-labeled actin filaments used in TIRF microscopy assays were prepared by copolymerizing unlabeled actin monomers and rhodamine-labeled actin monomers (Cytoskeleton catalog # AR05) at a 7:3 molar ratio (total actin concentration 1 µM) at room temperature for 2 hr in the dark. For optical trapping/confocal microscopy assays, 10% biotinylated actin filaments were prepared by copolymerizing purified unlabeled actin monomers and biotinylated actin monomers (Cytoskeleton catalog # AB07) at a 9:1 molar ratio (total actin concentration 1 µM) at room temperature for 2 hr. The biotinylated F-actin was subsequently stabilized and fluorescently labeled by adding 1.2 molar equivalents of Alexa Fluor 555 Phalloidin (ThermoFisher catalog # A34055).

## Expression cloning

Vectors for expression of WT *H. sapiens* vinculin ABD (879–1066), *H. sapiens* metavinculin ABD (879–1134), *H. sapiens* α-catenin ABD (αE-catenin$_{664-906}$), α-catenin ABDΔC (αE-catenin$_{664-871}$), vinculin ABD-NCSwap (αE-catenin$_{664-708}$-vinculin$_{916-1041}$-αE-catenin$_{837-906}$), and vinculin ABD-NCSwapΔC (αE-catenin$_{664-708}$-vinculin$_{916-1041}$-αE-catenin$_{837-871}$) were constructed by inserting the corresponding cDNA sequence into a pET bacterial expression vector with an N-terminal His6-tag, Strep-tag, TEV cleavage site, and, for proteins used in fluorescence imaging experiments, a subsequent C-terminal Halo-tag by Gibson assembly (*Gibson et al., 2009*). Constructs for proteins used in Cryo-EM structural studies are not Halo-tagged. Three WT α-catenin TIRF trials were performed with a previously reported Halo-tagged α-catenin ABD construct (αE-catenin$_{671-906}$), and no difference was observed.

## Expression and purification of actin-binding proteins

C-terminally GFP-tagged mouse myosin V HMM and myosin VI S1 were purified from SF9 cells using published protocols (*Wang et al., 2000*). Human calmodulin (CaM) was purified from Rosetta2(DE3) *E. coli* cells using a published protocol (*Putkey et al., 1985*) and stored in gel filtration buffer (20 mM Tris-Cl pH 8.0, 100 mM NaCl, 2 mM β-mercaptoethanol) supplemented with 5% v/v glycerol.

All other ABPs were expressed in Rosetta2(DE3) *E. coli* cells (Novagen) grown in LB media at 37°C to an optical density of 0.8–1.0 and induced with 0.7 mM IPTG. After induction, the cells were grown for 16 hr at 16°C, then cell pellets were collected and stored at −80°C until use. To purify Halo-tagged ABPs for fluorescent labeling, cells pellets were resuspended in lysis buffer (50 mM Tris-Cl pH 8.0, 150 mM NaCl, 5% v/v glycerol, 2 mM β-mercaptoethanol, 20 mM imidazole) and lysed with an Avestin Emulsiflex C5 homogenizer, after which the lysate was clarified at 15,000 g for 30 min. Cleared lysate was incubated with Ni-NTA resin (Qiagen) for 1 hr on a rotator at 4°C, after which the flow-through was discarded and the resin was washed with five bed volumes of lysis buffer. Proteins were subsequently eluted in elution buffer (50 mM Tris-Cl pH 8.0, 150 mM NaCl, 5% v/v glycerol, 2 mM β-mercaptoethanol, 300 mM imidazole). Purified His-tagged TEV protease (prepared according to a published protocol [*Tropea et al., 2009*]) was added at 0.05 mg/mL working concentration, then the eluate was dialyzed against dialysis buffer (20 mM Tris-Cl pH 8.0, 300 mM NaCl, 5% v/v glycerol, 2 mM β-mercaptoethanol) for 16 hr. The protein solution was then reapplied to Ni-NTA resin, and the flow-through was collected.

Protein was then sequentially purified by a HiTrapQ HP anion exchange column (GE Healthcare) followed by size exclusion chromatography on a Superdex 200 Increase column in gel filtration buffer supplemented with 10% v/v glycerol, and then snap-frozen in liquid nitrogen and stored at −80°C. Non-Halo-tagged proteins prepared for cryo-EM studies were purified identically, except glycerol was omitted from all buffers.

After rapid thawing before use, proteins were clarified by ultracentrifugation at 45,000 rpm in a TLA100 rotor for 10 min at 4°C. All protein concentrations were estimated using the Bradford colorimetric assay (Pierce), calibrated with BSA.

## Synthesis of Halo-JF-646 and labeling of ABPs

Fluorescent dye JF-646 (*Grimm et al., 2015*) NHS-ester building block (TOCRIS) was conjugated with Halo-tag ligand amine O4 (Promega) by synthetic chemistry according to published protocols (*Grimm et al., 2017*). Briefly, 1.5 equivalents of amine O4 ligand were added to one equivalent of the JF-646 NHS-ester in DMF followed by adding 5% triethylamine. The reaction was vigorously stirred for 16 hr at room temperature and the product was purified by silica gel chromatography, dried by SpeedVac (ThermoFisher), and reconstituted in DMSO.

For optical trapping/confocal microscopy assays, HaloTag Alexa Fluor 488 Ligand (Promega) was utilized as described above, followed by desalting through a PD SpinTrap G-25 column (GE Healthcare) according to the manufacturer's protocol to remove unreacted dye before use. To label the Halo-tagged actin-binding proteins with Halo-JF-646 for TIRF microscopy assays, two equivalents of synthesized Halo-JF-646 dye was added to the protein solution, followed by incubation for at least 2 hr in the dark at 4°C before use. Subsequent removal of excess dye was not required, as JF-646 is a fluorogenic dye (*Grimm et al., 2015*).

## Correlative force spectroscopy and confocal microscopy assays

Experiments were performed at room temperature (approximately 25°C) on a LUMICKS C-Trap instrument combining confocal fluorescence with dual-trap optical tweezers (*Hashemi Shabestari et al., 2017*; *Wasserman et al., 2019*). The optical traps were cycled through pre-set positions in the five channels of a microfluidic flow cell by an automated stage (*Figure 1A*). Channels 1–3 were separated from each other by laminar flow, which we utilized to form actin filament tethers between two 4 µm-diameter streptavidin-coated polystyrene beads (Spherotech) held in optical traps with a stiffness of 0.3 pN/nm. We first captured a single bead in each trap in channel 1. The traps were then transferred to channel 2, containing 5–20 nM Alexa 555 phalloidin-stabilized, 10% biotinylated F-actin in motility buffer ('MB': 20 mM MOPS pH7.4, 5 mM $MgCl_2$, 0.1 mM EGTA, 1 mM DTT) supplemented with 1 µM dark phalloidin, where tethers were formed by briefly moving 1 of the two traps toward the other trap against the direction of flow, followed by rapidly moving the traps to channel 3, which contained only buffer (MB + 1 µM dark phalloidin). The presence of a tether was verified by carefully separating the traps and observing an associated increase in force when monitoring the force-extension curve, applying the minimum extension feasible to avoid prematurely rupturing the tether.

The traps were then moved to orthogonal channel 4 or 5, which contained 2 µM fluorescently labeled vinculin ABD or α-catenin ABD (diluted in MB supplemented with 1 µM dark phalloidin), respectively, and flow was ceased during data acquisition. Force data were acquired at 200 Hz during constant velocity (0.1 µm/s) pulling experiments while simultaneously acquiring 2-color confocal fluorescence scans at 33 ms line scan time, exciting Alexa Fluor 488 HaloTag ligand and Alexa Fluor 555 phalloidin fluorophores with laser lines at 488 nm and 532 nm, respectively.

Data analysis was performed using ImageJ and custom software provided by LUMICKS. Force data from the two traps were averaged and binned to the confocal frame interval. The intensity values are measured by drawing a box in ImageJ to measure the fluorescence intensities of actin and ABP in both channels with background subtraction, calculating the background from an equal-sized box from that frame in an area devoid of filaments or beads. For paired analysis, the 'high-force' and 'low-force' averages were calculated only from the final tether to rupture, as long as the entire force trace has at least 10 quantifiable confocal image frames. For fluorescence-force correlation plots, only single-filament tethers selected based on *Figure 1—figure supplement 2A–C* were used. $I_{ABP}$/$I_{Actin}$ values were normalized by dividing the values in each recording by the largest value observed during that recording. K-means clustering analysis was performed to identify the cutoff force in the correlation plots for each ABP. Briefly, the force-fluorescence data were grouped into two clusters using scikit-learn (*Pedregosa et al., 2011*). For both α-catenin and vinculin, the data separated along the force axis, providing a threshold force for each ABP. Furthermore, a silhouette analysis for both ABPs confirmed that the data should not be clustered into more than two clusters (*Kaufman and Rousseeuw, 2009*).

## TIRF force reconstitution assays

Glass coverslips (Rectangular: Corning 22 × 50 mm #1½ Cover Glass; Square: Fisherbrand 22 × 22 mm #1½ Microscope Cover Glass) were cleaned by 30 min 100% acetone wash, 10 min 100% ethanol wash, and 2 hr 2% Hellmanex III liquid cleaning concentrate (HellmaAnalytics) wash in a bath sonicator followed by rinsing with water. The cleaned glass coverslips were coated with 1 mg/mL mPEG5K-Silane (Sigma) in a 96% ethanol, 10 mM HCl solution for at least 16 hr. After coating, the coverslips were rinsed with 96% ethanol and water, then air-dried and stored at 4°C until use. Flow cells were prepared with one square and one rectangular coverslip, both coated with mPEG-Silane. Double-sided adhesive tape (3M) was used to make ~4 mm-wide flow chambers between the coverslips, which were open on both sides to facilitate buffer exchange when adding components for imaging.

For each assay, 6 mg/mL anti-GFP antibody (Sigma #G1546) solution reconstituted in water was first introduced into the flow chamber and incubated for 2 min. Subsequently, MB containing 0.075 µM GFP-myosin V S1 and 0.15 µM GFP-myosin VI S1 were flowed into the chamber and incubated for another 2 min. A solution of 1 mg/mL bovine serum albumin (BSA) in MB was then flowed into the flow chamber and incubated for at least 2 min, after which 1 µM rhodamine-labeled F-actin in MB was flowed into the chamber and incubated for 20–30 s. The flow chamber was rinsed with MB

buffer to remove F-actin not bound to the rigor-state motors, then imaging buffer (MB without ATP, supplemented with 0.01% Nonidet P-40 substitute [Roche], 1 µM calmodulin, 15 mM glucose [Sigma], 1 µg/mL glucose oxidase [Sigma], and 0.05 µg/mL catalase [Sigma]) containing 2 µM fluorescently labeled ABP was flowed into the chamber. The first movie (-ATP, no force) was then recorded. A second imaging buffer, identical to the first but now including 100 µM ATP, was then introduced into the same chamber, and a second movie (+ATP, with force) was recorded. For each solution that was introduced, complete buffer exchange was facilitated by applying a filter paper at the other end of the flow chamber while pipetting.

Dual-color TIRF image sequences (movies) were recorded at room temperature (approximately 25°C) using a Nikon TiE inverted microscope equipped with an H-TIRF module and an Agilent laser launch, driven by Nikon Elements software. Images were taken every 2 s with an Apo TIRF 60 × 1.49 NA objective (Nikon) on an Andor iXon EMCCD camera; Rhodamine and JF646 fluorophores were excited by laser lines at 561 nm and 640 nm, respectively.

## Filament region quantification

To quantify ABP association with individual 'filament regions' of TIRF movies, we developed a custom ImageJ (*Schneider et al., 2012*) plugin (*Figure 2—figure supplement 1A*) which features a graphical user interface (GUI). The plugin takes as input two movie files, the actin channel and the ABP channel from a dual-color TIRF experiment, as well as an adjustable set of parameters (set by default in the GUI to the optimized values used in this study).

To identify regions of interest (ROIs) in each frame, the actin channel image series was preprocessed (unsharp mask, median filter, rolling ball subtraction), binarized, then segmented into contiguous regions of pixels using the built-in ImageJ plugin 'Analyze Particles'. ROIs whose centroids were fewer than 30 pixels from the edge of the field-of-view were excluded from further analysis due to incompatibility with downstream background subtraction procedures. The ROIs were then tracked through the image series and sorted into filament regions (representing individual filaments or small groups of filaments) by shortest Euclidean distance between ROI centroids in neighboring frames, with a maximum distance cutoff of 24 pixels. ROIs that were not matched with a pre-existing filament region by this criterion (i.e. whose centroid was greater than 24 pixels away from any ROI in the previous frame) were considered to represent a newly appeared filament region. Although this may result in overcounting the absolute number of filaments (should this be of interest; we do not believe this caveat impacts the conclusions of the present study), we find this procedure reliably handles events such as filament breakages and desultory motion in a completely automated fashion. To account for poorly tethered filaments fluctuating in and out of the evanescent field, only filament regions detected in at least 10 consecutive frames were included in the analysis.

Intensity in both the actin channel and the ABP channel were then quantified for each region. For each channel, the local background for each filament region in each frame was calculated as the mean gray value of the pixels from a 60 by 60 pixel box centered on the region's centroid, excluding pixels belonging to the region itself or any other filament region detected in the frame. Background-subtracted mean gray values were then calculated, followed by the ratio of these values across all frames in which the filament region was detected and their average, which we here report as the overall $I_{ABP}/I_{Actin}$ for that filament region. The program then outputs all frame and average $I_{ABP}/I_{Actin}$ values of the tracked filaments sorted by filament number for analysis, as well as a file containing all tracked regions.

## Cryo-EM sample preparation and data collection

F-actin was polymerized in G-Mg and KMEI from 5 µM unlabeled actin monomers at room temperature for 1 hr and then diluted to 0.6 µM in KMEI before use. Purified metavinculin ABD (879–1134) was diluted in KMEI to 10 µM before use. After screening grids prepared with finely sampled ABP concentrations, we found diluting purified α-catenin ABD (664-906) to 20 µM in KMEI before use gave an optimal balance between filament decoration and bundling.

Immediately before sample preparation, CF-1.2/1.3-3Au 300-mesh gold C-flat holey carbon cryo-TEM grids (Protochips) were plasma cleaned with a Hydrogen/Oxygen mixture for 5 s in a Gatan Solarus. Actin (3 µL) was first applied to the grid in the humidified chamber of a Leica EM GP plunge freezer and incubated for 60 s at 25°C. Actin-binding protein (3 µL) was then applied and incubated

for 30 s. Solution (3 μL) was then removed and an additional 3 μL of the same actin-binding protein solution was applied. After an additional 30 s, 3 μL of solution was removed, then the grid was back-blotted for 5 s, plunge-frozen in ethane slush, and stored in liquid nitrogen until imaging.

Cryo-EM data were recorded on a Titan Krios (ThermoFisher/FEI) operated at 300 kV equipped with a Gatan K2 Summit camera. SerialEM (*Mastronarde, 2005*) was used for automated data collection. Movies were collected at a nominal magnification of 29,000X in super-resolution mode resulting in a calibrated pixel size of 1.03 Å/pixel (superresolution pixel size of 0.515 Å/pixel), over a defocus range of −1.5 to −3.5 μm; 40 frames were recorded over 10 s of exposure at a dose rate of 6 electrons per pixel per second (1.5 electrons per $Å^2$ per second) for a cumulative dose of 60 electrons per $Å^2$.

## Cryo-EM image processing

Unless otherwise noted, all image processing was performed within the RELION-3.0 package (*Zivanov et al., 2018*). Movie frames were aligned and summed with 2 × 2 binning using the MotionCor2 algorithm (*Zheng et al., 2017*) as implemented in RELION (*Zivanov et al., 2019*), utilizing subframe motion correction with 5 × 5 patches. Contrast transfer function (CTF) parameters were estimated from non-doseweighted summed images with CTFFIND4 (*Rohou and Grigorieff, 2015*). Bimodal angular searches around psi angle priors were utilized in all subsequent 2D and 3D alignment/classification procedures. Around 2000 segments were initially manually picked, extracted, and subjected to 2D classification to generate templates for auto-picking. Helical auto-picking was then performed utilizing a step-size of 3 asymmetric units with a 27 Å helical rise. Segments were extracted from dose-weighted (*Grant and Grigorieff, 2015*) sum images in 512 × 512 pixel boxes which were not further down-sampled, then a second round of 2D classification followed by auto-picking with featureful class averages was performed.

A total of 237,503 particles from 1708 images (for the metavinculin ABD-actin dataset) and 540,553 particles from 7317 images (for the α-catenin ABD-actin dataset) were then extracted and subjected to whole-dataset 2D classification (*Figure 3—figure supplement 1*) using a 200 Å tube diameter and 300 Å mask diameter. 234,703 segments from the metavinculin ABD-actin dataset and 428,335 particles from the α-catenin ABD-actin dataset contributed to featureful class averages and were selected for 3D analysis.

All subsequent 3D classification and 3D auto-refine steps were primed with estimates of helical rise and twist of 27.0 Å and −167.0°, respectively, utilizing an initial reference low-pass filtered to 35 Å resolution, with the outer tube diameter set to 200 Å, inner tube diameter set to −1, and the mask diameter set to 300 Å. The first round of 3D classification into three classes was performed utilizing a reconstruction of a bare actin filament (EMBD-7115) as the initial reference. A second iteration of 3D classification was then performed as above, utilizing a featureful class with clear ABP density produced by the first round as the initial reference. For both datasets, this second round of 3D classification yielded two classes with helical parameters similar to the initial estimates and well-resolved 3D features, and one junk class with aberrant helical parameters and distorted features (*Figure 3—figure supplement 1*). Segments contributing to the two good classes were then pooled (215,369 particles for the metavinculin ABD-actin dataset and 414,486 particles for the α-catenin ABD-actin dataset) for 3D auto-refinement.

The first round of auto-refinement was then performed using one of the two good 3D averages as an initial reference. All masks for subsequent post-processing steps were calculated with 0 pixel extension and a six pixel soft edge from the converged reconstruction produced by that round of refinement, low-pass filtered to 15 Å and thresholded to fully encompass the density map. The first-round post-processing was performed with a 50% z length mask, followed by CTF refinement without beam-tilt estimation and Bayesian polishing (*Zivanov et al., 2019*). A second round of auto-refinement was then performed using the converged reconstruction from the first round as the initial reference. The second-round post-processing was performed with a 30% z length mask, followed by a second round of CTF refinement with beam-tilt estimation and Bayesian polishing. A final round of auto-refinement was then performed using the converged reconstruction from the second round as the initial reference. We found that this iterative procedure of tightening the mask for polishing resulted in substantial resolution improvements, potentially by mitigating the effects of medium-range disorder in F-actin previously speculated to limit the resolution of reconstructions of this filament (*Galkin et al., 2012*; *Merino et al., 2018*).

The final reconstructions converged with helical rise of 27.1 Å and twist of −167.1° for the metavinculin ABD–F-actin complex, and a helical rise of 27.0 Å and twist of −166.9° for the α-catenin ABD-F-actin complex, consistent with our finding that actin rearrangements evoked by these ABPs are minimal (*Figure 5H*). Final post-processing was performed with a 30% z length mask, leading to global resolution estimates of 2.9 Å for the metavinculin ABD–F-actin complex and 3.2 Å for the α-catenin ABD–F-actin complex by the gold-standard Fourier shell correlation (FSC) 0.143 criterion (*Figure 3—figure supplement 1*). B-factors of both datasets estimated during post-processing were then used to generate sharpened, local-resolution filtered maps with RELION. The key statistics summarizing cryo-EM image processing are reported in *Table 1*.

Asymmetric focused classification (without alignment) utilizing masks isolating the ABD region showed no evidence of segments with unoccupied binding sites (data not shown), suggesting that decoration of actin filaments by both 10 µM metavinculin ABD and 20 µM α-catenin ABD was essentially complete, with 100% occupancy at the limit of detection of current methods.

## Model building and refinement

Sharpened, local-resolution filtered maps as described above were used for model building. The 2.9 Å and 3.2 Å density maps were of sufficient quality for de novo atomic model building. As structures of components were available, initial models of actin (PDB 3j8a), metavinculin ABD (PDB 3jbk) truncated to residues 981–1131 and α-catenin ABD (PDB 4igg chain B) truncated to residues 699–871 were fit into the density map using Rosetta (*Wang et al., 2016*). Models were subsequently inspected and adjusted with Coot (*Brown et al., 2015*; *Emsley et al., 2010*), and regions that underwent significant conformational rearrangements were manually rebuilt. The models were then subjected to several rounds of simulated annealing followed by real-space refinement in Phenix (*Adams et al., 2010*; *Afonine et al., 2018*) alternating with manual adjustment in Coot. A final round of real-space refinement was performed without simulated annealing. The key statistics summarizing model building, refinement, and validation are reported in *Table 1*.

## Molecular graphics and structure analysis

Structural figures and movies were prepared with ChimeraX (*Goddard et al., 2018*). Per-residue RMSD analysis was performed with UCSF Chimera (*Pettersen et al., 2004*) as previously described (*Zhang et al., 2015*). The surface area of actin-binding interfaces was calculated with PDBePISA (*Krissinel and Henrick, 2007*; (EMBL-EBI). Model quality was assessed with EMRinger (*Barad et al., 2015*) and MolProbity (*Chen et al., 2010*) as implemented in Phenix.

## Sequence alignments

Protein sequences of human vinculin (UniProt Accession Code P18206-2), human metavinculin (P18206-1), human αE-catenin (P35221), human αN-catenin (P26232), and human αT-catenin (Q9UI47) were aligned with ClustalOmega (*Sievers and Higgins, 2014*; EMBL-EBI).

## Quantification and statistical analysis

Plotting and statistical analysis of data from TIRF force reconstitution assays and force-spectroscopy/confocal microscopy assays was performed with GraphPad Prism 8. All the details can be found in the figure legends of these figures and in the Method details. The data collection and refinement statistics of the cryo-EM structures can be found in *Table 1*. Resolution estimations of cryo-EM density maps and statistical validation performed on the deposited models are described in the Method details.

## Data and code availability

The atomic coordinates for the metavinculin ABD–F-actin complex and α-catenin ABD–F-actin complex have been deposited in the Protein Data Bank (PDB) with accession codes 6UPW and 6UPV, and the corresponding cryo-EM density maps in the Electron Microscopy Data Bank (EMDB) with accession codes EMD-20844 and EMD-20843.

The code for analyzing TIRF movies is freely available as an ImageJ plugin with a graphical user interface at https://github.com/alushinlab/ActinEnrichment. (copy archived at https://github.com/elifesciences-publications/ActinEnrichment; *Alushinlab, 2020*).

All other data are available in the manuscript or supplementary materials.

## Acknowledgements

We gratefully acknowledge Pinar Gurel (RU) for initial efforts in establishing the force reconstitution TIRF assay, Luke Lavis (HHMI Janelia) for the gift of JF-646 for pilot studies, and Yasuharu Takagi and James Sellers (NHLBI DIR) for the gift of myosin proteins and training in their purification. We also thank Sara Tafoya and Jordi Cabanas-Danés (LUMICKS) for assistance with C-trap experiments and data analysis, Muzaddid Sarker and Sharon Campbell (UNC) for the gift of metavinculin ABD used in cryo-EM studies, Rui Gong (RU) for assistance with cryo-EM data collection and analysis, and Mark Ebrahim and Johanna Sotiris (RU Cryo-EM Resource Center) for assistance with cryo-EM data collection. Template cDNA for α-catenin constructs was obtained from David Rimm via AddGene (#24194). This work was funded by grants from the Irma T Hirschl/Monique Weill-Caulier Trust and Pew Charitable Trusts to GMA, and NIH High-Risk High-Reward grants to GMA (5DP5OD017885, Early Independence Award) and SL (1DP2HG010510, New Innovator Award).

## Additional information

### Funding

| Funder | Grant reference number | Author |
|---|---|---|
| Irma T. Hirschl Trust | Research Award | Gregory M Alushin |
| Pew Charitable Trusts | Pew Scholar Award | Gregory M Alushin |
| National Institutes of Health | 5DP5OD017885 | Gregory M Alushin |
| National Institutes of Health | 1DP2HG010510 | Shixin Liu |

The funders had no role in study design, data collection and interpretation, or the decision to submit the work for publication.

### Author contributions

Lin Mei, Conceptualization, Resources, Data curation, Formal analysis, Validation, Investigation, Visualization, Methodology, Writing - original draft, Writing - review and editing; Santiago Espinosa de los Reyes, Matthew J Reynolds, Software, Formal analysis, Methodology, Writing - review and editing; Rachel Leicher, Investigation, Methodology, Writing - review and editing; Shixin Liu, Formal analysis, Supervision, Funding acquisition, Methodology, Project administration, Writing - review and editing; Gregory M Alushin, Conceptualization, Data curation, Formal analysis, Supervision, Funding acquisition, Visualization, Methodology, Writing - original draft, Project administration, Writing - review and editing

### Author ORCIDs

Lin Mei (iD) http://orcid.org/0000-0002-5056-4547
Santiago Espinosa de los Reyes (iD) http://orcid.org/0000-0003-4510-8296
Matthew J Reynolds (iD) http://orcid.org/0000-0002-2501-9280
Shixin Liu (iD) http://orcid.org/0000-0003-4238-7066
Gregory M Alushin (iD) https://orcid.org/0000-0001-7250-4484

### Decision letter and Author response

Decision letter https://doi.org/10.7554/eLife.62514.sa1
Author response https://doi.org/10.7554/eLife.62514.sa2

## Additional files

### Supplementary files

• Transparent reporting form

## Data availability

The atomic coordinates for the metavinculin ABD-F-actin complex and α-catenin ABD-F-actin complex have been deposited in the Protein Data Bank (PDB) with accession codes 6UPW and 6UPV, and the corresponding cryo-EM density maps in the Electron Microscopy Data Bank (EMDB) with accession codes EMD-20844 and EMD-20843. The code for analyzing TIRF movies is freely available as an ImageJ plugin with a graphical user interface at https://github.com/alushinlab/ActinEnrichment (copy archived at https://github.com/elifesciences-publications/ActinEnrichment). All other data are available in the manuscript or supplementary materials.

The following datasets were generated:

| Author(s) | Year | Dataset title | Dataset URL | Database and Identifier |
| --- | --- | --- | --- | --- |
| Mei L, Alushin GM | 2020 | Metavinculin ABD-F-actin complex | https://www.rcsb.org/structure/6UPW | RCSB Protein Data Bank, 6UPW |
| Mei L, Alushin GM | 2020 | Alpha-E-catenin ABD-F-actin complex | https://www.rcsb.org/structure/6UPV | RCSB Protein Data Bank, 6UPV |

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
