## [Decision Letter]

**Acceptance summary:**

The actin cytoskeleton has the unique ability to propagate mechanical stimuli at cell-cell and cell-matrix interactions, resulting in the maturation of adhesion complexes. The actin filaments that are placed under tension in these regions create unique binding sites for actin-binding proteins. Vinculin and α-catenin have been shown differentially bind to tensed and relaxed actin filaments, but the mechanism remains elusive. Here, Mei et al. address this important question. They combine beautiful cryo-EM structures of actin decorated by metavinculin or α-catenin with optical trap and TIRF motility assays to explore the how actin binding proteins interact with actin under tension.

**Decision letter after peer review:**

[Editors’ note: the authors submitted for reconsideration following the decision after peer review. What follows is the decision letter after the first round of review.]

Thank you for submitting your work entitled "Molecular mechanism for direct actin force-sensing by α-catenin" for consideration by *eLife*. Your article has been reviewed by three peer reviewers, one of whom is a member of our Board of Reviewing Editors, and the evaluation has been overseen by a Senior Editor. The reviewers have opted to remain anonymous.

Our decision has been reached after consultation between the reviewers. Based on these discussions and the individual reviews below, we regret to inform you that your work will not be considered further for publication in *eLife* at this stage. As you will see from the comments below the reviewers were impressed by your cryoEM structures and felt the work was of general interest and potentially suitable for e*Life*. However they had a number of concerns particularly concerning the force experiments.

The reviewers all agree that your work has the potential to make a stronger paper. You will need to make the decision what you want to do with the manuscript, especially given the current situation. However, if you think you can solve most of the issues raised then we encourage you to resubmit to e*Life* later in the year.

Reviewer #1:

The actin cytoskeleton has the unique ability to propagate mechanical stimuli at cell-cell and cell-matrix interactions, resulting in the maturation of adhesion complexes. The actin filaments that are placed under tension in these regions create unique binding sites for actin-binding proteins. Vinculin and α-catenin have been shown differentially to bind to tensed and relaxed actin filaments, but the mechanism remains elusive. Here, Mei et al. explore the binding of vinculin and α-catenin to actin under tension to address this important question.

The authors first develop an assay to pull single actin filaments and assess the binding intensity of vinculin and α-catenin. Vinculin did not show differential binding to filaments under high or low pulling force, but α-catenin showed a mild preference for filaments pulled beyond 1 pN. To validate this result, the authors developed another assay where opposite polarity myosin motors were used to place actin filaments under tension. This showed that α-catenin binding was enhanced when ATP was added, whereas vinculin binding was unchanged. The authors obtain beautiful cryo-EM structures of actin decorated by metavinculin or α-catenin at high resolution. The major binding site for the two proteins on actin was largely similar, whereas the minor binding site was distinct. In particular, the C-terminal extensions followed different paths on actin, and the extreme 35 amino acids on α-catenin were unresolved. The authors speculate that these unresolved residues mediated the force-sensitivity. Truncating these residues abolished force sensitivity, but switching the major binding region of α-catenin with vinculin did not, suggesting that the C-terminal extension of α-catenin is necessary and sufficient to mediate force-sensitivity. While this work provides important insight into how forces can be propagated at tensed cellular environments, I have a few major concerns that should be addressed by the authors.

1) In Figure 1, high force was taken to be the 5 frames before breakage and low force to be 5 frames before that. Therefore, a specific amount of time is taken to represent high versus low force, which is not appropriate. Furthermore, the low number of replicates raises questions on the robustness of the result. Therefore, I find that 1D and E weaken the author's claims, which are better represented in 1F and G, where the data is plotted against force. In the latter, however, it is not clear why 1 pN is chosen to divide the data. A regression analysis is more appropriate, with R2 values reported.

2) A necessary assumption in the myosin assay is that myosins behave similarly in all experiments. However, this is not validated and may not be the case. For example, Video 7 (ABD-NCSwap) clearly has more bundled actin filaments after ATP addition than Video 6 (ABD-ΔC) and Video 8 (ABD-NCSwapΔC). The authors even hint that myosin inhibition can occur in certain cases. Additional experiments to assess myosin activity is required for these data to be reliable. A suitable assay would be myosin gliding in the presence of the different constructs, for example.

3) Metavinculin is used for structural studies to prevent bundling, and the authors validate its use by stating that it is similarly force-insensitive in the myosin assay. However, this is represented by a single experiment in Figure 2—figure supplement 1D, while it is clear from Figure 2—figure supplement 1B that there is significant variability in this type of data. Therefore, the use of metavinculin as a proxy for vinculin in structural analyses is not properly justified. Additional replicates will be required.

4) In testing whether the D-loop interactions contribute to force-sensitivity, mutations are made in α-catenin to resemble metavinculin at the D-loop interface. The mutations abolish binding to actin altogether, and an interpretation provided by the authors is that "this interface does not have a separable role in force-activated actin binding". This interpretation cannot be correct if no binding was detected altogether, and the authors should remove the statement.

5) The motivation to truncate the disordered region of the C-terminal extension of α-catenin is not entirely clear. The residues of the C-terminal extension preceding this disordered stretch are clearly different between α-catenin and vinculin in the two cryo-EM structures. This region is therefore also a good candidate force sensor. The authors need to explain why they didn't switch this region between the two proteins.

Reviewer #2:

Using distinct biophysical assays, Mei et al. investigate the effect of pN scale forces on the interaction between actin filaments and two adhesion-associated actin-binding proteins (either α-catenin or its homolog vinculin). They show that forces moderately promote the binding of α-catenin but not vinculin. Extending their previous structural work on (meta) vinculin, they then determine high-resolution structures of the ABDs of either protein on actin filaments in the absence of force, which reveal differences in the C-terminal tail regions. Using chimeric constructs, they provide additional biochemical support for the C-terminus of α-catenin conferring force sensitivity.

Overall, the paper clearly contains several strong elements that merit publication. It focuses on a pertinent question of broad interest (the molecular mechanisms of force sensing in the actin cytoskeleton) and the data quality is high throughout. The TIRFM motility assay is a useful addition to the field and the observation of force-sensitive binding of α-catenin is in itself novel and interesting.

The central weakness of the paper is that it falls rather short in providing answers to the most pressing questions: The nature of the force-induced conformational change in actin filaments remains ultimately unclear. Similarly, the "force-sensing" C-terminal element of α-catenin (residues 872-906) is not resolved in the cryoEM structure. Hence, the detailed mechanism of force sensing remains to be elucidated.

That being said, I still feel that the paper deserves to be published even in the absence of a comprehensive structural mechanism. My main points rather address some of the immediate technical and conceptual issues that require improvement.

1) Both vinculin and α-catenin ABDs are known to bundle actin filaments. I was therefore surprised to see that the authors seem to neglect potential confounding effects that bundling might cause in their reconstitutions. Both TIRFM and optical trapping experiments are done in a configuration in which filaments appear to be clearly bundled. The authors even mention that filaments become more strongly bundled in the TIRFM assay upon ATP addition. I am concerned that some of their observations might be equally explainable by differential effects of bundling/filament overlap configurations on vinculin and α-catenin. The authors need to perform control experiments under conditions where they clearly observe and analyze only individual actin filaments to demonstrate that the effect of force is indeed based on a conformational change in the filament itself.

2) The authors should be more explicit and conscious about the ABD concentrations employed in their in vitro assays. This information can currently only be found in the Materials and methods section (should be part of the figures/captions!). Why did they chose to work at 2uM for either protein in buffers of low ionic strength? Previous measurements show that both α-catenin and vinculin ABDs binds with sub-micromolar affinity to filaments (Hansen et al., 2013 and Le Clainche et al., 2010), which means that the authors work at or close to saturation. This should mask or at least severely diminish the effects that force might exert on ABD binding. In my opinion, the authors need to take much greater care of the fractional occupancy of filaments by ABDs (by measuring the ABD affinities) under their experimental conditions to ensure that they work still in a sensitive concentration regime. Determining exact affinities in the presence and absence of force in the TIRFM assay also ought to give them additional information concerning the relative number of binding sites at saturation (maximal signal), which might allow them to discriminate between the proposed force sensing mechanisms.

3) The "tug of war" motility assay employed by the authors, while ingenious, suffers from a central weakness. The opposite polarity motors are equally likely to produce tensile AND compressive forces on the surface-bound filament. This important limitation is not discussed by the authors at all. Admittedly, the optical trapping assay shows that the effect can also be observed for tensile forces only. However, the magnitude in the TIRFM assay is much greater, which is why one cannot help but wonder whether compression is the true binding-promoting cue. I am fully aware that this cannot easily be addressed experimentally. However, the authors should be more open in discussing this alternative interpretation.

4) The "steric exclusion" model put forward by the authors posits that adjacent α-catenin ABDs interfere with synchronous binding. How can this be reconciled with the observation of cooperative α-catenin ABD binding (Hansen et al., 2013), indicating stabilizing effects? The authors cite and superficially discuss this point, but ultimately do not provide a satisfactory explanation. Is there any indication of partial binding in their cryoEM data? Along the same vein, have they compared the binding stoichiometry of α-catenin (either wt or C-terminal deletion mutants) at saturation via TIRFM or co-sedimentation?

Reviewer #3:

Mei et al. present a very interesting study of force-dependent binding of actin-binding domains to filamentous actin. Using optical trapping, they demonstrate forces > 1pN are sufficient to show enhanced actin binding of a-catenin, a process they call force-activated binding, but not vinculin. The authors identify a C-terminal motif of a-catenin that is required for force-activated binding and that confers similar behavior to vinculin when added. Using cryoEM, they obtain structures of both a-catenin and vinculin bound to actin, showing where the C-terminal motif binds. This work provides interesting new information about how forces on filaments can influence actin-binding proteins. However, the mechanistic insight is limited by the lack of data on how a 1pN force alters actin and how a-catenin might sense that change. In addition, I have concerns about interpretation of the optical trapping and TIRF data and am unclear why constructs and labeling schemes vary across the different experimental methods used.

1) The optical trapping results appear to pool data on single filaments and bundled filaments (as noted in the Materials and methods), which makes interpretation of the data very difficult and raises questions about the conclusions.

a) How many single filaments vs bundled filaments are included in the data set? The caption of Figure 1 indicates that it shows data on individual filaments, but the images of actin do not look like single filaments. For example, in the series of images of actin filaments in Figure 1C, the 4th actin filament looks like it could be several filaments, and the 5th image seems to show a broken filament. I'm not convinced that a single rupture force conclusively shows that only one filament is being imaged, as all filaments need not be load bearing and may only be laterally associated with a filament that is attached to the bead. Could a calibrated measurement of filament fluorescence intensity be used to quantify the number of filaments in a bundle?

b) It seems that the amount of force felt by each filament will depend on precisely how each is coupled to the beads and to each other, which is hard to know. Won't the inclusion of actin bundles lead to an error in estimating the force at which binding of a-catenin increases?

c) How do the force-extension curves obtained in this study compare with previous measurements of actin force-extension, such as Liu and Pollack (Biophys J, 2002)? This might help to identify single vs bundled filaments.

d). How do the authors correct for filament length in their analysis? The effect of 1pN on a fluctuating filament will depend on the length of the filament.

e) In the vinculin ABD binding experiment in Figure 1B, why is the relative ABD intensity varying so much (~30%) if binding of the ABD has equilibrated on the filament? And how is the intensity ratio normalized in Figure 1F and 1G?

2) The myosin TIRF experiments also raise several questions:

a) Are the filaments imaged in the TIRF experiments single filaments or bundles (or both)? As discussed above, can calibrated fluorescence intensity be used to identify single vs bundled filaments?

b) Won't some filaments in the myosin TIRF assay experience compression while others experience tension? What about twist? How does this affect data interpretation and the force that is responsible for a-catenin force-activated binding?

c) The authors note the variability in the assay. Any thoughts on the source of this variability? Would a lower density of filaments in Figure 2B and 2C enable easier data collection? Also, the variability in the TIRF data shown in Figure 6—figure supplement 1 is puzzling. Why are some values of normalized intensity negative?

3) The cryoEM data is very impressive. While the structures of a-catenin and vinculin ABDs on actin are well done, they don't provide insight into how force-activated binding might be happening.

a) While imaging filaments under a load in cryoEM would be very difficult, can the authors identify two different classes of filaments in their data, analogous to their "low force" and "high force" grouping of trapping data in Figure 1? If a force of only 1pN is necessary to drive enhanced binding, perhaps filaments under forces that are above and below that may be captured during freezing, which could provide valuable mechanistic insight.

b) Was a structure of the vinculin ABD with the a-catenin C-terminal motif solved to confirm that it interacts with the filament in the same way as it does on a-catenin?

4) The choice of filament labeling strategies and ABD used appears to vary from experiment and would benefit from more explanation and/or experiments with the same constructs.

a) In the cryoEM structures, metavinculin was used rather than vinculin to avoid bundling. Why wasn't metavinculin used in the optical trapping and TIRF experiments, given that bundling complicates analysis and interpretation of the data?

b) The optical trapping experiments used phalloidin-labeled actin filaments, while the TIRF experiments used rhodamine-labeled actin filaments. Why weren't the experiments done with the same actin labeling method?

5) The proposed models seem reasonable but would benefit from more consideration of fluctuations and comparison with other existing models.

a) It is unclear how the thermal fluctuations of a filament under physiological conditions would affect the proposed a-catenin steric exclusion and actin filament conformational change models. Won't the small changes associated with a force of 1pN already be sampled by thermal fluctuations? While detailed modeling is beyond the scope of the manuscript, some discussion of thermal fluctuations and the proposed models would be helpful.

b) The authors cite Ishiyama et al., 2018, which presents a different mechanism for a-catenin force dependence based on conformational unfolding of the ABD under force, but there is little discussion of it. Are the proposed models compatible with this?

c) As a minor point, the authors say that they report the first direct observation of enhanced actin binding by an actin-binding protein. However, Hayakawa et al., 2011, which the authors cite, previously reported tension-dependent binding of cofilin to actin.

[Editors’ note: further revisions were suggested prior to acceptance, as described below.]

Thank you for submitting your article "Molecular mechanism for direct actin force-sensing by α-catenin" for consideration by *eLife*. Your article has been reviewed by two peer reviewers, one of whom is a member of our Board of Reviewing Editors, and the evaluation has been overseen by Suzanne Pfeffer as the Senior Editor. The reviewers have opted to remain anonymous.

The reviewers have discussed the reviews with one another and the Reviewing Editor has drafted this decision to help you prepare a revised submission.

Summary:

The actin cytoskeleton has the unique ability to propagate mechanical stimuli at cell-cell and cell-matrix interactions, resulting in the maturation of adhesion complexes. The actin filaments that are placed under tension in these regions create unique binding sites for actin-binding proteins. Vinculin and α-catenin have been shown differentially bind to tensed and relaxed actin filaments, but the mechanism remains elusive. Here, Mei et al. address this important question. They combine beautiful cryo-EM structures of actin decorated by metavinculin or α-catenin with optical trap and TIRF motility assays to explore the how actin binding proteins interact with actin under tension.

The manuscript is much improved and the additional replicate experiments and modified analysis of optical tweezer data strengthen the results. The varied experimental approaches used by the authors are impressive.

The reviewers still have concerns about the TIRF motility data (listed below). Their suggestion is to either address these directly now, or to discuss in full the caveats to the TIRF experiments and explain how they will address them in future work.

1) The ABD concentrations employed in the TIRFM experiments remain a source of concern. Instead of resorting to speculations about whether they might not be at saturation, the reviewers feels the authors should try to perform experiments to find out. A recent preprint by the Weis and Hanein labs shows that the presence/absence of the N-terminal H0 can drastically affect a-catenin affinity. Given that a-catenin ABD intensities are close to background levels in TIRFM, the use of a shorter and high-affinity construct might be helpful.

2) The authors opted to not repeat the TIRF motility experiments using single filament densities as requested. While the single motor control experiments (Figure 2—figure supplement 2) are in principle suitable controls, it is concerning to see so little specific filament binding of a-catenin in the raw data. As detailed also in the previous comments by reviewer 3, the authors are encouraged to invest some time in optimizing their TIRFM assay to perform these important controls.

3) The authors should comment about their findings in relation to the study from Weis' lab.

---

## [Author Response]

[Editors’ note: the authors resubmitted a revised version of the paper for consideration. What follows is the authors’ response to the first round of review.]

Reviewer #1:The actin cytoskeleton has the unique ability to propagate mechanical stimuli at cell-cell and cell-matrix interactions, resulting in the maturation of adhesion complexes. The actin filaments that are placed under tension in these regions create unique binding sites for actin-binding proteins. Vinculin and α-catenin have been shown differentially to bind to tensed and relaxed actin filaments, but the mechanism remains elusive. Here, Mei et al. explore the binding of vinculin and α-catenin to actin under tension to address this important question.The authors first develop an assay to pull single actin filaments and assess the binding intensity of vinculin and α-catenin. Vinculin did not show differential binding to filaments under high or low pulling force, but α-catenin showed a mild preference for filaments pulled beyond 1 pN. To validate this result, the authors developed another assay where opposite polarity myosin motors were used to place actin filaments under tension. This showed that α-catenin binding was enhanced when ATP was added, whereas vinculin binding was unchanged. The authors obtain beautiful cryo-EM structures of actin decorated by metavinculin or α-catenin at high resolution. The major binding site for the two proteins on actin was largely similar, whereas the minor binding site was distinct. In particular, the C-terminal extensions followed different paths on actin, and the extreme 35 amino acids on α-catenin were unresolved. The authors speculate that these unresolved residues mediated the force-sensitivity. Truncating these residues abolished force sensitivity, but switching the major binding region of α-catenin with vinculin did not, suggesting that the C-terminal extension of α-catenin is necessary and sufficient to mediate force-sensitivity. While this work provides important insight into how forces can be propagated at tensed cellular environments, I have a few major concerns that should be addressed by the authors.1) In Figure 1, high force was taken to be the 5 frames before breakage and low force to be 5 frames before that. Therefore, a specific amount of time is taken to represent high versus low force, which is not appropriate. Furthermore, the low number of replicates raises questions on the robustness of the result. Therefore, I find that 1D and E weaken the author's claims, which are better represented in 1F and G, where the data is plotted against force. In the latter, however, it is not clear why 1 pN is chosen to divide the data. A regression analysis is more appropriate, with R2 values reported.

We appreciate the reviewer’s concerns with the low-force / high-force analysis. We have significantly revised the optical trapping portion of our manuscript, notably including substantially more trials. Out of 183 optical trapping traces, only 13 fit our criteria for low-force / high-force analysis, and only 21 fit our criteria for single-filament analysis.

As the reviewer surely appreciates, tether-to-tether variability is a major limitation of these experiments. We thus respectively disagree that the single tether analysis weakens our claims; rather, we believe it is important to show that we do see an increase in a-catenin binding along individual tethers when force is applied. We do believe the 5 frame window is necessary to see this effect, based on the noise level of the data (Figure 1B, C). We believe our findings from this analysis justifies the subsequent analysis based on pooling multiple tethers.

Based on the comments of the other two reviewers, we have implemented a detailed procedure for ensuring that the force vs. binding analysis focuses on tethers composed of individual actin filaments. Thus, while our overall dataset contains 183 recordings, only 21 tethers were included in this analysis based on the stringent criteria we imposed.

Although it is possible that determining a suitable fit function would be possible if we had many more points, to our eyes binding transitions from force-uncorrelated to strong binding in an essentially step-like fashion, within the noise level of the data (see Figure 1B and 1C: we now state this explicitly in the text). Therefore, regression analysis is not a feasible choice. Regardless, the reviewer’s point that 1 pN is a subjectively determined threshold is well-taken. We have performed K-means clustering analysis to more objectively identify the cutoff force value directly from the data in both plots, which we describe in the Results and the Materials and methods section. We have since used these force values, as well as the original 1 pN criteria, to divide the data in Figure 1F and Figure 1G. We observe a significant difference for a-catenin, but not vinculin, in both cases, suggesting this analysis is not sensitive to chosen threshold force.

Finally, we hope the reviewer can appreciate the low-throughput nature and low efficiency of these experiments. Out of 183 recordings, only 21 tethers could be included in the stringent single-filament subset, a “yield” of 11%. Under the current working conditions imposed by the ongoing COVID-19 pandemic with social distancing restrictions, our additional data was acquired with substantial difficulty. It is currently not feasible to obtain the massive datasets that would likely be required for a meaningful, well justified regression analysis as proposed by the reviewer.

2) A necessary assumption in the myosin assay is that myosins behave similarly in all experiments. However, this is not validated and may not be the case. For example, Video 7 (ABD-NCSwap) clearly has more bundled actin filaments after ATP addition than Video 6 (ABD-ΔC) and Video 8 (ABD-NCSwapΔC). The authors even hint that myosin inhibition can occur in certain cases. Additional experiments to assess myosin activity is required for these data to be reliable. A suitable assay would be myosin gliding in the presence of the different constructs, for example.

We thank the reviewer for this suggestion. We have performed single-motor experiments with the wild-type constructs. As noted by the all the reviewers, significant actin bundling occurs with both constructs in the dual motor experiments. We find that significant bundling also occurs in the presence of either the a-catenin ABD or the vinculin ABD with either myosin V alone or myosin VI alone, which unsurprisingly disrupts processive motility. This result is not surprising as any kind of actin movement, driven by either one myosin motor or two myosin motors, will help actin filaments sample larger space and cause bundling in the presence of an actin-bundling protein. Note that we are confident the motors are active (we always perform control gliding filament assays with both individual motors in the absence of ABPs where we do observe processive motility, data not shown). Therefore, a detailed quantification of the impacts of these constructs on myosin activity (e.g. processive gliding) is not possible, and we did not extend this analysis to all of the constructs employed in our work, which would be a substantial effort with limited potential for meaningful interpretation.

However, fortuitously we find that neither the a-catenin ABD nor the vinculin ABD displays ATP-enhanced binding in the presence of either motor alone, despite the presence of bundling. This strongly suggests that the increased a-catenin binding is due to the tug-of-war between motors of opposing directionality, rather than simply being an indirect effect of bundling. We have thus included this data in the revised manuscript (Figure 2—figure supplement 2) which we believe substantially strengthens our conclusions.

3) Metavinculin is used for structural studies to prevent bundling, and the authors validate its use by stating that it is similarly force-insensitive in the myosin assay. However, this is represented by a single experiment in Figure 2—figure supplement 1D, while it is clear from Figure 2—figure supplement 1B that there is significant variability in this type of data. Therefore, the use of metavinculin as a proxy for vinculin in structural analyses is not properly justified. Additional replicates will be required.

As requested, we have performed additional replicates, which support our conclusion that metavinculin is force insensitive. This data is included in Figure 2—figure supplement 1D, supporting our use of metavinculin in the structural experiments. We believe use of metavinculin is also supported by the predictive power of the structures in delineating the force detector of a-catenin.

4) In testing whether the D-loop interactions contribute to force-sensitivity, mutations are made in α-catenin to resemble metavinculin at the D-loop interface. The mutations abolish binding to actin altogether, and an interpretation provided by the authors is that "this interface does not have a separable role in force-activated actin binding". This interpretation cannot be correct if no binding was detected altogether, and the authors should remove the statement.

The reviewer’s point is well-taken. We have adjusted this statement to: “While these data suggest that the α-catenin D-loop interactions contribute to overall affinity for F-actin, the complete lack of binding is refractory to determining whether this interface has a separable role in force-activated actin binding.”

5) The motivation to truncate the disordered region of the C-terminal extension of α-catenin is not entirely clear. The residues of the C-terminal extension preceding this disordered stretch are clearly different between α-catenin and vinculin in the two cryo-EM structures. This region is therefore also a good candidate force sensor. The authors need to explain why they didn't switch this region between the two proteins.

We thank the reviewer’s comment on this issue. The protein residues we have switched in order to make the chimera, a-catenin residues 837-871, does include this region. The reason why we did not truncate this region is because a previous study has clearly shown that truncating this region of a-catenin will abolish its actin binding (Pokutta et al., 2002). As we have discussed in the main text this is consistent with our structure in which we have found this region directly forms a binding interface with actin.

Reviewer #2:Using distinct biophysical assays, Mei et al. investigate the effect of pN scale forces on the interaction between actin filaments and two adhesion-associated actin-binding proteins (either α-catenin or its homolog vinculin). They show that forces moderately promote the binding of α-catenin but not vinculin. Extending their previous structural work on (meta) vinculin, they then determine high-resolution structures of the ABDs of either protein on actin filaments in the absence of force, which reveal differences in the C-terminal tail regions. Using chimeric constructs, they provide additional biochemical support for the C-terminus of α-catenin conferring force sensitivity.Overall, the paper clearly contains several strong elements that merit publication. It focuses on a pertinent question of broad interest (the molecular mechanisms of force sensing in the actin cytoskeleton) and the data quality is high throughout. The TIRFM motility assay is a useful addition to the field and the observation of force-sensitive binding of α-catenin is in itself novel and interesting.The central weakness of the paper is that it falls rather short in providing answers to the most pressing questions: The nature of the force-induced conformational change in actin filaments remains ultimately unclear. Similarly, the "force-sensing" C-terminal element of α-catenin (residues 872-906) is not resolved in the cryoEM structure. Hence, the detailed mechanism of force sensing remains to be elucidated.That being said, I still feel that the paper deserves to be published even in the absence of a comprehensive structural mechanism. My main points rather address some of the immediate technical and conceptual issues that require improvement.

We of course agree with the reviewer that visualizing force-evoked structural rearrangements in F-actin is the critical next step in dissecting the mechanism of force-activated binding, as we clearly state in the Discussion. However, as this effectively requires developing a new structural biology method, we reaffirm the reviewer’s point that this is not a reasonable request for revision of our current paper. The current manuscript is not data-light, and we believe it provides a substantial advance that sets the foundation for pursuing a structure of a-catenin bound to F-actin in the presence of force as a follow-up study.

1) Both vinculin and α-catenin ABDs are known to bundle actin filaments. I was therefore surprised to see that the authors seem to neglect potential confounding effects that bundling might cause in their reconstitutions. Both TIRFM and optical trapping experiments are done in a configuration in which filaments appear to be clearly bundled. The authors even mention that filaments become more strongly bundled in the TIRFM assay upon ATP addition. I am concerned that some of their observations might be equally explainable by differential effects of bundling/filament overlap configurations on vinculin and α-catenin. The authors need to perform control experiments under conditions where they clearly observe and analyze only individual actin filaments to demonstrate that the effect of force is indeed based on a conformational change in the filament itself.

The reviewer’s point is well-taken. We have acquired substantially larger optical trapping datasets for both vinculin and a-catenin and implemented a detailed procedure to focus our force vs. binding analysis on tethers composed of individual actin filaments (Figure 1F, 1G). This analysis still clearly shows a force-dependent binding increase only for a-catenin.

Additionally, at the recommendation of reviewer 1, we have performed TIRFM reconstitution experiments in the presence of individual myosin motor proteins (myosin V alone and myosin VI alone). Fortuitously, these experiments also show increased bundling in the presence of ATP, but no evidence of increased a-catenin binding (Figure 2—figure supplement 2, Video 9-12). This strongly suggests that increased binding in the dual motor TIRFM experiments is not simply due to bundling.

We believe that collectively these new experiments convincingly make the case that force activated a-catenin binding occurs at the level individual filaments, and have substantially strengthened our paper. Furthermore, we believe demonstrating that force-activated binding also occurs in context of actin bundles is a strength, not a weakness, since this is the cytoskeletal architecture recognized by a-catenin in vivo, a point which we have emphasized in the text.

2) The authors should be more explicit and conscious about the ABD concentrations employed in their in vitro assays. This information can currently only be found in the Materials and methods section (should be part of the figures/captions!). Why did they chose to work at 2uM for either protein in buffers of low ionic strength? Previous measurements show that both α-catenin and vinculin ABDs binds with sub-micromolar affinity to filaments (Hansen et al., 2013 and Le Clainche et al., 2010), which means that the authors work at or close to saturation. This should mask or at least severely diminish the effects that force might exert on ABD binding. In my opinion, the authors need to take much greater care of the fractional occupancy of filaments by ABDs (by measuring the ABD affinities) under their experimental conditions to ensure that they work still in a sensitive concentration regime. Determining exact affinities in the presence and absence of force in the TIRFM assay also ought to give them additional information concerning the relative number of binding sites at saturation (maximal signal), which might allow them to discriminate between the proposed force sensing mechanisms.

We have now included the ABD concentrations in the figure legends. However, due to substantial variability in reported F-actin binding KDs in the literature, we believe the order of magnitude of the concentration is far more relevant than the exact value. For example, Rimm et al. reported in their 1995 PNAS paper that the a-catenin ABD has a 3μM affinity to F-actin. We chose 2 μm because based on numerous previous reports, including the ones by Hansen et al. and Rimm et al., both proteins have essentially μM affinity for F-actin. 2 μm, in our hands, is the lowest concentration we can see significant effects for the force-sensitive actin binding for a-catenin.

Hansen et al. reported approximately 0.5 μM F-actin affinity for the a-catenin ABD. Note that we observe very little a-catenin binding in the absence of force in the TIRFM experiments (Figure 2C), explicitly contradicting the reviewer’s assertion that we are working close to saturation. Hansen et al. noted distinct behaviors of N-terminally labelled GFP a-catenin ABD and unlabeled a-catenin ABD, so a potential source of the apparent lower affinity of a-catenin in our hands is the presence of a C-terminal halo tag. It is also plausible that there are discrepancies in our measurements of protein concentration. We used a Bradford assay calibrated with BSA (as noted in the Materials and methods), while Hansen et al. did not report their method of protein concentration measurement.

Le Clainche et al. performed co-sedimentation studies to examine the affinity of the vinculin ABD for F-actin (Figure 4D of their paper). Although it is difficult to exactly compare cosedimentation assays and our TIRFM studies due to the many differences in the conditions (immobilization of F-actin, etc.), at the point in their curves most closely related to our assays (2 μM vinculin ABD and 1 μM actin), they report that ~25 % of the vinculin ABD is bound, i.e. 0.5 μM. As the maximum stoichiometry of vinculin ABD to actin protomers is 1:1, this would correspond to 50% of sites being occupied, precisely the range where we would expect to see sensitivity to additional modulating factors of binding such as force.

Although it is once again difficult to compare assays with different modalities, we found it required an order of magnitude higher concentrations of both ABDs (10 μM for metavinculin and 20 μM for a-catenin) to achieve saturation in our cryo-EM studies. While we find it is common that higher concentrations are required to achieve saturation in cryo-EM of ABP-Factin complexes, this nevertheless suggests that the concentrations employed in our biophysical experiments are not nearing saturation.

Finally, we appreciate the reviewer’s suggestion that it would be useful to explicitly determine the effects of force on binding affinity by performing our TIRFM experiments at different concentrations. As noted in the text, there is considerable variability between replicates even when the assay is performed with identical concentrations. We thus do not believe it would be feasible to perform these experiments unless the TIRFM assay is systematically improved, which we believe is an appropriate subject for a subsequent follow-up study, which we now note in the Discussion.

In summary, we believe that we are not working close to saturation, and that performing systematic studies of the quantitative impact of force on ABD affinity are not currently feasible as a revision to our present manuscript. We were very careful in our writing to make clear that, although we quantitate our data to support our conclusions, the fundamental phenomenon we report is simply an “increase” in F-actin binding by the a-catenin ABD in the presence of force. A quantitative dissection of Kd vs. force would constitute a study in itself, which would require substantial refinement of the methods we introduce in this paper.

3) The "tug of war" motility assay employed by the authors, while ingenious, suffers from a central weakness. The opposite polarity motors are equally likely to produce tensile AND compressive forces on the surface-bound filament. This important limitation is not discussed by the authors at all. Admittedly, the optical trapping assay shows that the effect can also be observed for tensile forces only. However, the magnitude in the TIRFM assay is much greater, which is why one cannot help but wonder whether compression is the true binding-promoting cue. I am fully aware that this cannot easily be addressed experimentally. However, the authors should be more open in discussing this alternative interpretation.

We are certainly aware of the multiple types of forces that can in principle be generated in our dual-motor assay. We believe it is unlikely that compression is the dominant binding promoting cue, since many tethers in our optical trapping experiments are bent at the onset of the experiment (e.g. Figure 1B), suggesting they are compressed beyond their resting contour length when captured in the microfluidic flow cell, but do not show increased a-catenin binding. Regardless, we have included a sentence about the different motor forces generated in the TIRFM assay, including torsional forces, to the Results section. Dissecting these different forces through elaborations and improvements to the assay will be an important subject for future studies.

4) The "steric exclusion" model put forward by the authors posits that adjacent α-catenin ABDs interfere with synchronous binding. How can this be reconciled with the observation of cooperative α-catenin ABD binding (Hansen et al., 2013), indicating stabilizing effects? The authors cite and superficially discuss this point, but ultimately do not provide a satisfactory explanation. Is there any indication of partial binding in their cryoEM data? Along the same vein, have they compared the binding stoichiometry of α-catenin (either wt or C-terminal deletion mutants) at saturation via TIRFM or co-sedimentation?

We thank the reviewer for the insightful comment. We believe these two theories, our “steric exclusion” model and the cooperative binding of α-catenin, are not mutually exclusive. It is plausible that an α-catenin molecule can have both cooperative and inhibitory effects for a neighboring α-catenin molecule to bind actin: the cooperativity can facilitate its engaging, while the steric exclusion can reduce it. For α-catenin’s actin binding interaction, having binding cooperativity does not rule out the possibility that there may still be steric hindrance between neighboring molecules globally reducing binding, which can be alleviated by force on actin. Indeed, these opposing interaction modes provide multiple axes for regulation which could be harnessed by the cell. We have added an expanded discussion about this issue in the main text (subsection “Molecular mechanism for detecting force on actin through a flexible structural element”).

We are confident that actin filaments are fully decorated in our Cryo-EM data. Our lab has extensive experience in analyzing heterogeneity in cytoskeletal complexes, including occupancy (e.g. Kim et al., 2016). As indicated in the Materials and methods, we do not see any evidence of partial occupancy in the cryo-EM data through classification methods, and we believe if there was partial binding we would have found it. Our structure clearly shows that, at saturation, the binding stoichiometry is 1:1, which we now explicitly emphasize. There may be some confusion as the a-catenin binding site spans two actin protomers, but keep in mind that each actin protomer also contacts 2 a-catenin molecules.

As the binding stoichiometry of the wild-type is 1:1 with the actin protomer (e.g. the maximum possible), it is not plausible that the C-terminal deletion mutant would have a higher binding stoichiometry. We thus do not believe it is meaningful to examine binding stoichiometry through other methods, since it is clearly maximal for the wild-type as indicated from the structural data.

Reviewer #3:Mei et al. present a very interesting study of force-dependent binding of actin-binding domains to filamentous actin. Using optical trapping, they demonstrate forces > 1pN are sufficient to show enhanced actin binding of a-catenin, a process they call force-activated binding, but not vinculin. The authors identify a C-terminal motif of a-catenin that is required for force-activated binding and that confers similar behavior to vinculin when added. Using cryoEM, they obtain structures of both a-catenin and vinculin bound to actin, showing where the C-terminal motif binds. This work provides interesting new information about how forces on filaments can influence actin-binding proteins. However, the mechanistic insight is limited by the lack of data on how a 1pN force alters actin and how a-catenin might sense that change. In addition, I have concerns about interpretation of the optical trapping and TIRF data and am unclear why constructs and labeling schemes vary across the different experimental methods used.

We thank the reviewer for their overall appreciation of the significance of our findings. While we of course agree that a structure of a-catenin bound to F-actin in the presence of force will be very insightful, this requires development of a novel structural biology method, and thus goes significantly beyond the scope of a revision. Beyond identifying a-catenin as a force activated F-actin binding protein (itself a significant advance), we nevertheless believe that our current work provides significant insights into the mechanism, notably the identification of the force-detector through our combined cryo-EM structural studies and functional biophysics approach. Beyond constraining plausible biophysical mechanisms for future studies, this finding immediately enables cell biological experiments to investigate the role of a-catenin’s force activated binding in vivo by examination of the DC mutant. Thus, we believe the significance of our work will be broadly appreciated by the readership of *eLife*.

1) The optical trapping results appear to pool data on single filaments and bundled filaments (as noted in the Materials and methods), which makes interpretation of the data very difficult and raises questions about the conclusions.a) How many single filaments vs bundled filaments are included in the data set? The caption of Figure 1 indicates that it shows data on individual filaments, but the images of actin do not look like single filaments. For example, in the series of images of actin filaments in Figure 1C, the 4th actin filament looks like it could be several filaments, and the 5th image seems to show a broken filament. I'm not convinced that a single rupture force conclusively shows that only one filament is being imaged, as all filaments need not be load bearing and may only be laterally associated with a filament that is attached to the bead. Could a calibrated measurement of filament fluorescence intensity be used to quantify the number of filaments in a bundle?

We very much appreciate this point, which is similar to ones brought up by reviewers 1 and 2. Broadly, we have collected substantially expanded optical trapping datasets, which enabled us to institute a strenuous series of criteria for excluding multi-filament tethers, described in the text and Figure 1—figure supplement 2, for the force vs. fluorescence analysis displayed in Figure 1F-G. To specifically address the case of laterally associated filaments which are not loadbearing: Motivated by the fact that such filaments are unlikely to be exactly the same length as the load-bearing filament, we examined fluorescence line-scans of the actin intensity. These show a clear step-like drop in intensity when such filaments are present (Figure 1—figure supplement 2C), enabling us to exclude them from further analysis.

b) It seems that the amount of force felt by each filament will depend on precisely how each is coupled to the beads and to each other, which is hard to know. Won't the inclusion of actin bundles lead to an error in estimating the force at which binding of a-catenin increases?

The reviewer’s point is well-taken, and we now only include single-filament tethers in the force vs. binding analysis in Figure 1F-G. Additionally, we have implemented an objective clustering based analysis to identify the threshold force for the binding increase, as discussed above in our answer to the reviewer #1.

c) How do the force-extension curves obtained in this study compare with previous measurements of actin force-extension, such as Liu and Pollack (Biophys J, 2002)? This might help to identify single vs bundled filaments.

We appreciate this suggestion. The rupture forces reported by Liu and Pollack with cantilevers for single filaments were generally > 200 pN, i.e. 2 orders of magnitude greater than the modal rupture force for single filaments that we now report (6 pN), with tethers surviving several cycles of stretch. While it is possible that the different apparatuses employed (including our dual-color imaging), and differences in loading rate and/or tethering chemistry, can give rise to this discrepancy, and we do not want to cast aspersions on the pioneering work of Liu and Pollack, it is unclear how these authors ensured they were pulling on single filaments. We believe that the very large discrepancy makes it difficult to directly compare the force extension curves.

Two of our co-authors (M. Reynolds and S. Liu) recently developed a novel analysis procedure for analyzing DNA force-extension curves (Leicher…Liu. Biorxiv, 2019) to estimate persistence length. We attempted to apply this analysis to our data, but did not observe a consistent difference between clear multi-filament tethers and single filament tethers, likely due to deviations from the WLC model for actin, the low rupture forces, as well as the generally short length of our recordings for single filaments, limiting power of the fits.

We thus believe that, while imperfect, examination of the breaking force distribution is the best analysis for distinguishing individual filaments, when coupled with our other criteria for excluding multi-filament tethers.

d) How do the authors correct for filament length in their analysis? The effect of 1pN on a fluctuating filament will depend on the length of the filament.

We observe no systematic relationship between the force dependent increase in binding (the difference between high-force and low-force averages) and filament length, or breaking force and filament length (see Author response image 1). We thus have not corrected for filament length in our analysis.

**Author response image 1. sa2fig1:** 

e) In the vinculin ABD binding experiment in Figure 1B, why is the relative ABD intensity varying so much (~30%) if binding of the ABD has equilibrated on the filament? And how is the intensity ratio normalized in Figure 1F and 1G?

We believe the intensity is fluctuating due to the variations inherent to a single-molecule pulling experiment. Sources of variation include fluctuations in focus, fluctuations in binding density along the tether, performance of the fluorophores, and performance of the detector. We believe the I_ABD_/I_actin_ analysis is the best possible correction for these factors, yet variations do remain.

The intensity ratio in Figure 1F-G is normalized on a tether-by-tether basis by the largest value in the image sequence for each tether, which is stated in the Materials and methods.

2) The myosin TIRF experiments also raise several questions:a) Are the filaments imaged in the TIRF experiments single filaments or bundles (or both)? As discussed above, can calibrated fluorescence intensity be used to identify single vs bundled filaments?

It is likely that both single filaments and bundles are present in the TIRF data, which we explicitly note in the text. At the reviewer’s suggestion, we analyzed the distribution of raw actin intensity values of all filament regions in a representative video. If were able to distinguish bundles from individual filaments, we would anticipate observing a multi-modal distribution, where each mode represents the number of filaments in a bundle. Instead, we observe a unimodal distribution (see the plot in Author response image 2), suggesting that we cannot discriminate the number of filaments in each region from intensity alone. We nevertheless believe the bundles are physiologically relevant: indeed, actomyosin bundles are the primary substrate of a-catenin and vinculin in vivo. We thus believe force-activated binding to bundles is an important finding, which we emphasize in the text.

A point of concern raised by the other reviewers is that bundling increases upon activating the myosins, which is correct. At the suggestion of reviewer 1, we have now conducted singlemotor experiments (Figure 2—figure supplement 2), which, while still showing an ATP-dependent increase in bundling, show no increase in actin binding by either ABD. We believe these experiments convincingly demonstrate that force-activated binding is not simply a product of increased bundling in the myosin-motor reconstitution experiments.

b) Won't some filaments in the myosin TIRF assay experience compression while others experience tension? What about twist? How does this affect data interpretation and the force that is responsible for a-catenin force-activated binding?

We agree with the reviewer that tension, compression, and torque may all be present in the TIRF assays. We have thus carefully stated that “myosin generated forces” are sufficient to activate a-catenin binding. While the optical-trapping experiments suggest tension is sufficient to activate binding, we have now included an explicit discussion of this issue in the text.

c) The authors note the variability in the assay. Any thoughts on the source of this variability? Would a lower density of filaments in Figure 2B and 2C enable easier data collection? Also, the variability in the TIRF data shown in Figure 6—figure supplement 1 is puzzling. Why are some values of normalized intensity negative?

We would be delighted to conclusively know the source of variability in these experiments, as we would then hopefully be able to eliminate it. Our current best guess is batch-to-batch variation in our cover-glass preparation procedure, which strongly affects the amount of background present in our experiments. While we do perform a background subtraction step for quantification, systematic variations may nevertheless affect the I_ABD_ / I_actin_ distribution. We now explicitly include a discussion of this issue in the Results section.

Obtaining an appropriate filament density is a delicate balance, as too low of a filament density tends to result in the filaments being quickly shattered by the motors, precluding analysis. Too many filaments prevent the reliable identification of individual regions. The actin concentration we employed was optimized in light of these considerations.

Negative values are expected to occasionally arise due to Poisson distributed shot-noise on the camera. In filament regions featuring very low binding (essentially zero), it is formally possible that the local background will randomly contain slightly more counts than the region itself, producing a negative value for the I_ABD_ upon background subtraction. The I_actin_ will always have a positive value, since actin intensity is used for thresh-holding and region identification. Therefore, I_ABD_/I_actin_ can occasionally have a (small) negative value, which is indeed what we observe.

3) The cryoEM data is very impressive. While the structures of a-catenin and vinculin ABDs on actin are well done, they don't provide insight into how force-activated binding might be happening.

Respectfully, we strongly disagree with this statement. The cryo-EM structures explicitly enabled us to precisely identify the force detector in a-catenin. Furthermore, our design of the vinculin-a-catenin chimera which gains force-activated binding was enabled by the structures, which provides perhaps the strongest evidence in our paper that the a-catenin C-terminus is the force detector.

a) While imaging filaments under a load in cryoEM would be very difficult, can the authors identify two different classes of filaments in their data, analogous to their "low force" and "high force" grouping of trapping data in Figure 1? If a force of only 1pN is necessary to drive enhanced binding, perhaps filaments under forces that are above and below that may be captured during freezing, which could provide valuable mechanistic insight.

We observe no evidence of this type of variability in our dataset, as described in extensive detail in the Materials and methods, as well as Figure 3—figure supplement 1. As our lab has extensive expertise in analyzing heterogenous filament datasets (e.g. Kim et al., 2016), we believe if it was there, we would have found it.

We realize that there is literature suggesting that the conditions of cryo-EM sample preparation expose actin filaments to tensile forces, which we cite in the Introduction. It is our opinion that the nature of these forces remains speculative, and we are not comfortable building upon this speculation in our work without well-controlled experiments. We are currently pursuing such studies, which are of sufficient scope to be the subject of an independent follow-up paper.

b) Was a structure of the vinculin ABD with the a-catenin C-terminal motif solved to confirm that it interacts with the filament in the same way as it does on a-catenin?

We did not obtain such a structure, which would be a massive investment of time and resources which are currently quite limited during the ongoing COVID 19 pandemic. We believe such a request is outside the scope of what is normally considered appropriate for a revision. While there is formally a very small chance that this chimera adopts a different binding mode, this is far from the most parsimonious explanation for our data.

4) The choice of filament labeling strategies and ABD used appears to vary from experiment and would benefit from more explanation and/or experiments with the same constructs.a) In the cryoEM structures, metavinculin was used rather than vinculin to avoid bundling. Why wasn't metavinculin used in the optical trapping and TIRF experiments, given that bundling complicates analysis and interpretation of the data?

As the reviewer might expect, the studies described in our paper evolved over the course of several years. As vinculin is the canonical isoform expressed in most cells, we chose to employ this for our initial studies. We then turned to structural studies to interrogate the mechanism, where we chose to focus on metavinculin to avoid bundling. As this structure had substantial predictive power when we engineered vinculin mutants, we believe this choice was well-justified.

b) The optical trapping experiments used phalloidin-labeled actin filaments, while the TIRF experiments used rhodamine-labeled actin filaments. Why weren't the experiments done with the same actin labeling method?

Although presented in the manuscript with the optical trapping experiments first, we actually performed the dual-motor TIRF experiments with the wild-type constructs first during the course of our studies. At the time, we were concerned that phalloidin (which stabilizes F-actin) would prevent force-dependent conformational changes in F-actin recognized by mechanosensitive ABPs. We thus chose to employ rhodamine-labeled actin. Once we discovered a-catenin’s force-activated actin binding activity, we then pursued the optical trapping studies to validate this observation and dissect the contribution of tension from other motor-dependent phenomena (compression and torsion, as noted by the reviewer).

As the optical trapping experiments had to be performed at very low F-actin concentrations in order to have a chance at capturing individual filaments, they necessitated the use of phalloidin. Fortuitously, we found that phalloidin did not inhibit force-activated binding. We have also performed control experiments where we visually confirmed that un-labelled phalloidin does not impact force-activated binding to rhodamine-labelled filaments in myosin TIRF reconstitution experiments (data not shown).

We believe that demonstrating a-catenin’s force-activated binding to be independent of the actin-labelling strategy employed is a strength, not a weakness, of our manuscript.

5) The proposed models seem reasonable but would benefit from more consideration of fluctuations and comparison with other existing models.a) It is unclear how the thermal fluctuations of a filament under physiological conditions would affect the proposed a-catenin steric exclusion and actin filament conformational change models. Won't the small changes associated with a force of 1pN already be sampled by thermal fluctuations? While detailed modeling is beyond the scope of the manuscript, some discussion of thermal fluctuations and the proposed models would be helpful.

We thank the reviewer for this suggestion. We agree that low piconewton forces affecting the distribution of actin conformations is a reasonable and attractive model. We believe either mechanism we propose could operate in this framework, where discrete conformations are substituted with conformational ensembles. We have added an explicit note to this regard in the revised Discussion, including referencing previous work suggesting this regulatory mechanism for cofilin.

b) The authors cite Ishiyama et al., 2018, which presents a different mechanism for a-catenin force dependence based on conformational unfolding of the ABD under force, but there is little discussion of it. Are the proposed models compatible with this?

The mechanism proposed by Ishiyama et al. concerns force-dependent rearrangements in the N-terminus of the ABD (H0-H1), specifically force-dependent unfolding. Our actin-bound structure does indeed show this region unfolds upon actin-binding, which could reasonably be stabilized by force. This is a mechanism that could explain catch-bond formation (we have previously proposed a similar mechanism to explain vinculin catch-bond formation with F-actin in a commentary: Swaminathan, Alushin and Waterman, 2017), but it cannot explain force-activated binding. This is because the entire ABD would have to be bearing tension in order for this region, which does not directly contact actin, to “feel” force. As our paper focuses on force-activated binding rather than catch-bond formation, and already features a lengthy discussion, we did not delve into this in detail. However, this mechanism (which we emphasize to be quite speculative) is discussed at length in the competing pre-print from Volkman, Hanein, Weis, and colleagues which also reports a (somewhat lower resolution) structure of the a-catenin ABD bound to F-actin, that has emerged while our paper is under review (and that we now cite). We believe substantial further work (e.g. visualizing the entire cadherin complex bound to F-actin in the presence of tension) will be required to establish the mechanism of a-catenin catch-bonding to F-actin in detail.

c) As a minor point, the authors say that they report the first direct observation of enhanced actin binding by an actin-binding protein. However, Hayakawa et al., 2011, which the authors cite, previously reported tension-dependent binding of cofilin to actin.

We thank the reviewer for pointing out this oversight. We have included a statement that tension dependent binding of cofilin to F-actin was previously reported by Hayakawa et al., and also called into question by a more recent paper from Wioland, Jegou and Romet-Lemonne, 2019. We have also systematically pruned our paper of excessive claims of novelty, which we do not believe impacts the significance of our findings.

[Editors’ note: what follows is the authors’ response to the second round of review.]

Revisions for this paper:The manuscript is much improved and the additional replicate experiments and modified analysis of optical tweezer data strengthen the results. The varied experimental approaches used by the authors are impressive.The reviewers still have concerns about the TIRF motility data (listed below). Their suggestion is to either address these directly now, or to discuss in full the caveats to the TIRF experiments and explain how they will address them in future work.

Due to the related work which has very recently appeared from Hanein, Volkman, Weis, and colleagues (Xu et al., 10.7554/*eLife*.60878), also in e*Life*, we have elected to discuss these caveats of the TIRF studies and propose specific experiments to address them in future work. This has been included in the revised manuscript.

1) The ABD concentrations employed in the TIRFM experiments remain a source of concern. Instead of resorting to speculations about whether they might not be at saturation, the reviewers feels the authors should try to perform experiments to find out. A recent preprint by the Weis and Hanein labs shows that the presence/absence of the N-terminal H0 can drastically affect a-catenin affinity. Given that a-catenin ABD intensities are close to background levels in TIRFM (see below), the use of a shorter and high-affinity construct might be helpful.

We have emphasized that Xu et al. report a series of high-affinity N-terminal deletion constructs in the text, including a construct which lacks both H0 and H1, which has ~18 fold higher affinity for actin. We agree with the reviewer that examining a higher affinity construct, should it retain force-activated binding to actin, will likely be very useful for making accurate estimates of binding stoichiometry and their impact on force-activated binding. We will pursue these experiments as an immediate follow-up. The discussion of this caveat and our plan for addressing it are included in the revised manuscript.

2) The authors opted to not repeat the TIRF motility experiments using single filament densities as requested. While the single motor control experiments (Figure 2—figure supplement 2) are in principle suitable controls, it is concerning to see so little specific filament binding of a-catenin in the raw data. As detailed also in the previous comments by reviewer 3, the authors are encouraged to invest some time in optimizing their TIRFM assay to perform these important controls.

While our optical trapping studies support force-activated binding to individual filaments, we agree that it will be interesting to establish whether myosin motor activity is also sufficient to activate binding to individual filaments in addition to bundles. It may be feasible to achieve sufficiently low filament densities to avoid bundling in the presence of phalloidin, and we will pursue these experiments as an immediate follow-up. The discussion of this caveat and our plan for addressing it are included in the revised manuscript.

The outcome of these experiments (both points 1 and 2) will be posted as a bioRxiv pre-print and, if suitable, submitted as an e*Life* Research Advance. A statement to this effect has also been included in the revised manuscript.

3) The authors should comment about their findings in relation to the study from Weis' lab.

We have added detailed comparisons to the study of Xu et al., noting specific points of concordance and discordance between our papers in the Results section. This includes:

1) A quantitative comparison of the structural models and comments about their similarity.

2) Referencing the W859A mutant of Xu et al., which reduces actin affinity, in support of the importance of the tryptophan latch.

3) Noting the apparently conflicting results of a CTE deletion (in our hands, increased overall affinity and loss of force-activated binding in TIRF assays; in their hands, slight reduction in affinity in solution co-sedimentation assays), and potential reasons for this discrepancy.

We have furthermore included an additional paragraph in the Discussion regarding the catch-bonding model proposed by Xu et al., in relation to our present and previous work.